# A Multidisciplinary Approach for a Better Knowledge of the Benthic Habitat and Community Distribution in the Central and Western English Channel

Jean-Claude Dauvin [1,*], Jean-Philippe Pezy [1], Emmanuel Poizot [2,3], Sophie Lozach [4] and Alain Trentesaux [5]

1    Laboratoire Morphodynamique Continentale et Côtière, UMR CNRS 6143 M2C, Normandie University, UNICAEN, 2-4 rue des Tilleuls, 14000 Caen, France
2    Laboratoire Universitaire des Sciences Appliquées de Cherbourg, EA, Normandie University, UNICAEN 4253, 14000 Cherbourg, France
3    Conservatoire National des Arts et Métiers, Institut National des Sciences et Techniques de la Mer (CNAM/INTECHMER), B.P. 324, 50103 Cherbourg, France
4    The Centre for Environment, Fisheries & Aquaculture Science (Cefas), Lowestoft Laboratory, Pakefield Road, Lowestoft, Suffolk NR33 OHT, UK
5    Laboratoire d'Océanologie et de Géosciences, University of Lille, CNRS, Université Littoral Côte d'Opale, UMR 8187, LOG, 59000 Lille, France
*    Correspondence: jean-claude.dauvin@unicaen.fr

**Abstract:** About 80% of the seabed of the English Channel (EC) is covered by coarse sediment, from coarse sand to pebbles. Quantitative data on the benthic macrofauna in these types of sediment remains are rare due to the difficulty of using grab corers in such hard substrates. The deepest central part of the EC (45–101 m depth) was prospected during two VIDEOCHARM surveys in June 2010 and June 2011 to increase knowledge of such sublittoral coarse sediment benthic habitats. Sampling focussed on a longitudinal transect in the deepest part of the EC (13 boxes), extending from the western approach to the Greenwich meridian. Both indirect (side scan sonar, Remote Operated Vehicle) and direct (grab sampling with benthos determination, and grain-size analyses) approaches were used and combined, permitting description of the benthic habitats and communities using seven methods. Five benthic EUNIS habitats (European Nature Information System) were reported: MC3215, MD3211, MC4, MC3212 and MC4215, of which two extended main habitats (MC3211 and M23212) corresponded to an eastern/western gradient from sandy gravel to sandy gravel and pebbles sediment. Three other spatially discrete habitats were associated with poor coarse sand and gravel habitats as well as sandy gravel and pebbles with the presence of the brittle star *Ophiothrix fragilis*. Taxonomic richness of both extended habitats was on the same order of magnitude as the coarse sand habitat reported elsewhere in the EC, whilst the abundances were among the lowest in deeper areas with low nutrient input and low primary production. The epifauna appeared relatively homogenous in this type of sediment at the scale of the sampling area and was not determined to assign a EUNIS habitat/class. ROV footage illustrated the presence of large epifauna and provided valuable information to ground truth in other sampling methods such as side scan sonar mosaic. Grab photos showing surface sediment was relevant to determine the sediment type, whilst granulometric analyses gave additional information on fine particles content (typically very low).

**Keywords:** English Channel; sublittoral; benthic habitats; EUNIS classification; community distribution; multidisciplinary approach

## 1. Introduction

In the framework of the CHARM (Channel Habitat Atlas for Marine Resource management) project, the role of the macrobenthos as a fundamental prey resource for demersal fish had been determined across the three phases of this European project [1–4]. Additionally,

the description of benthic communities in marine coastal protected areas, including distribution and functioning of sensitive benthic habitats, remains an important challenge [5,6]. Historical works at the scale of the English Channel (EC) as a model of a megatidal sea have been undertaken in the 1960s and 1970s, respectively, by two teams, Holme's coming from the Plymouth laboratory (United Kingdom) [7,8], and Cabioch's from the Roscoff Biological Station (France) [9–11]. Back then, benthic sampling was mainly qualitative, i.e., using a sampling technique such as the 'Rallier du Baty' dredge by the French team, and was devoted to the description of spatial distribution of the main benthic communities and to identify benthic species. During these studies, the distance between sampling stations only permitted explaining two large distribution patterns: first, the role of hydrodynamics in the sediment spatial distribution, with fine sediment located in areas with low tidal currents and, conversely, the absence of fine sediment in areas with high currents reaching up to 5 knots in some parts of the Normand Breton Gulf, the Cotentin and the Dover Straits, where rocky outcrops occur locally; second, the presence of a climatic gradient from the western approach of the Channel, influenced by the Atlantic waters, to the eastern approach, influenced by the input of freshwater coming mainly from the Seine estuary, where the winter temperatures are lower than in the western basin. The result is the impoverishment of benthic species from the western approach, the richest, to the eastern approach, the poorest. Cabioch et al. [10] hence described the importance of the edaphic–climatic gradients for the benthic species distribution. Moreover, the authors in [7,8] described the presence of Sarnian species occurring in the hydrological isolated Normand–Breton Gulf. Holme [8] was the first to collect quantitative data for the macrobenthic communities in some soft-bottom communities at the scale of the whole EC. He described the difficulties of sampling coarse sand, gravel and pebbles using quantitative sampling gears such as grabs. Most quantitative data were then collected near the shore in sandy and muddy sediments, which were much easier to collect with the type of grabs available at the time.

Obtaining quantitative data represented a challenge in a major part of the EC [12–14]. Quantitative data were therefore missing in offshore benthic habitats of the EC dominated by coarse sediment (>80% of its surface). Supplementary data were hence needed to describe the structure and distribution of main benthic communities of EC and to study benthic ecosystem function. Following European benthic classification, i.e., EUNIS classification, the authors of [15] used sonar and video footage to study the diversity of marine benthic habitats. The authors of [16] used photos of the seabed to determine the benthic assemblages of the central EC (south of the Isle of Wight) and offshore of North Brittany. Following these projects, CEFAS promoted the development of such integrated approaches along the English side of the Channel, especially to study the mixed soft-hard bottom organisation of benthic communities and to produce habitat maps of the seabed in coastal UK waters [17–22].

To increase our knowledge of the benthic habitats in the central deeper areas of the EC, two VIDEOCHARM surveys were carried out in 2010 and 2011. The study area covered a profile across the EC and was designed in the continuity of the CHARM II sampling grid to expand existing databases and to complement the UK work carried out in English waters [11,23–26]. During the surveys, benthic habitats were studied using acoustic remote sensing techniques, coupled with in situ grab sampling and collections of video footage using a small remotely operated vehicle (ROV), for a total of seven methods.

The aims of this paper were: (1) to characterise the benthic habitats along the VIDEOCHARM profile using the seven available methods; (2) to provide quantitative macrobenthic data in this offshore area of the EC; and (3) to propose EUNIS classification of the benthic habitats and communities in the central and eastern parts of the EC.

## 2. General Characteristics of the English Channel

The EC is a shallow epicontinental sea (77,000 km$^2$) bordered by the United Kingdom and France located at the transition between the Atlantic Ocean and the North Sea [27–29]. The depth is about 100 m at the entrance of the Channel to the west, reaching 174 m in

its central trench then diminishes to 40 m in the Dover Strait [30]. The distribution of superficial sediment [31] and benthic communities [7–9,11,28,32] are the result of a tidal circulation, the offshore-inshore hydrodynamics leading to a bio-sedimentary gradient with extended pebble and sessile benthic communities dominating areas of strong tidal currents offshore, and fine sand and muddy fine sand communities inhabiting weak tidal currents areas (e.g., bays and estuaries) and the climatic conditions with thermic amplitude between minimal and maximal sea bottom temperature higher in the eastern basin than in the western basin. In the central part of the EC, the environmental conditions show a high hydrodynamic area offshore the Cotentin peninsula mainly covered by pebbles and rocky substrates, transitioning eastwards and westwards as residual currents decrease, to sediment dominated by gravel and coarse sand [30,31]. Following the benthic studies of Holme [7–11] completed a habitat mapping survey over an extensive area in the central part Channel using acoustic, underwater photography and traditional grab sampling techniques to create full-coverage modelled maps of the biotopes according to the EUNIS habitat classification system. The previous interpretations made by Holme and Cabioch for the central EC were very consistent with their own interpretations.

## 3. Material and Methods Used during the Videocharm 2011 and 2011 Campaigns

A series of 13 rectangular boxes was sampled during both the VIDEOCHARM 2010 (4–14 June) (boxes 1 to 7) and 2011 (14–29 June) (boxes 8 to 13) surveys combining side scan sonar (Table 1), Hamon grab sampling and video footage with a small ROV SeaBotix. All the boxes were south of the France–UK bordering to stay in French waters and were distributed from the Greenwich meridian to the EC entry, offshore Brest (Figure 1). Boxes 7–11 are located near the French coast due to the bad weather conditions in 2011; nevertheless, they were all on the circalittoral coarse sediment of the western part of the EC.

**Table 1.** Total length of side scan sonar profiles over each box of the two VIDEOCHARM surveys with the percentage of the identified morphological structures and the presence of dredge or trawl trace and wreck.

| Morphological Structures | Box | | | | | | | | | | | | |
|---|---|---|---|---|---|---|---|---|---|---|---|---|---|
| N° | 1 | 2 | 3 | 4 | 5 | 6 | 7 | 8 | 9 | 10 | 11 | 12 | 13 |
| Length (km) | 83.2 | 77.5 | 84.5 | 82.1 | 83.5 | 75.3 | 39.4 | 53.4 | 50.3 | 53.0 | 51.5 | 50.1 | 45.0 |
| Ribbons (km$^2$) | 0 | - | 0 | 0 | - | - | - | - | - | - | - | - | 0 |
| Furrows (km$^2$) | 0 | - | 0.6 | 2.56 | - | - | - | - | - | - | - | - | 0 |
| Small and medium dunes (km$^2$) | 0 | - | 0 | 0 | - | - | - | - | - | - | - | - | 0 |
| Very large dunes (km$^2$) | 0 | - | 1.6 | 1.3 | - | - | - | - | - | - | - | - | 16.4 |
| Sand veneer on rocks | 0 | - | 0 | 0 | - | - | - | - | - | - | - | - | 0 |
| Homogeneous zone (km$^2$) | 0 | - | 0 | 0 | - | - | - | - | - | - | - | - | 0 |
| Table rocks, outcropping or sub-flush rocks (km$^2$) | 0 | - | 2.6 | 3.26 | - | - | - | - | - | - | - | - | 0 |
| Rocky area (km$^2$) | 0 | - | 0.5 | 4.55 | - | - | - | - | - | - | - | - | 0 |
| Dredge or trawl trace (km) | 88.5 | - | 3.0 | 1.0 | - | - | - | - | - | - | - | - | 3.14 |
| Wreck or anthropic marks | 25 | - | 64 | 33 | - | - | - | - | - | - | - | - | 0 |

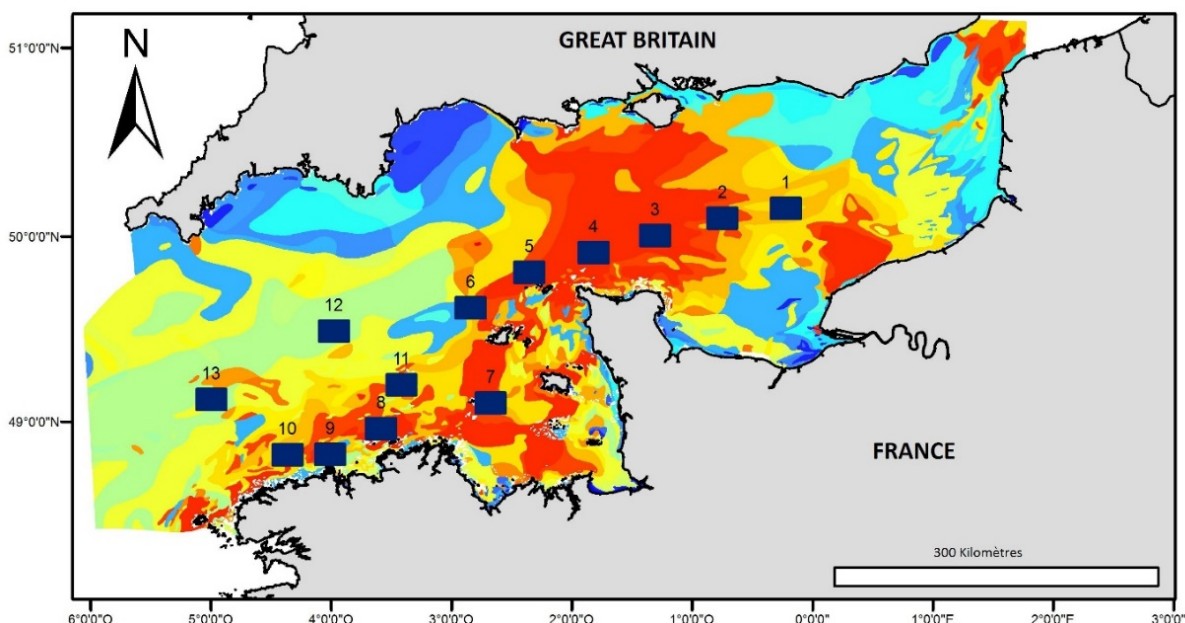

**Figure 1.** Location of the 13 boxes (1 to 13, blue rectangle) sampled during the VIDEOCHARM surveys in June 2010 and June 2011 with the map of the three main superficial sediment types in the English Channel: orange, pebbles and large gravel; yellow: gravel and blue: sands and muds (from 31 in 29).

　　　Seven methods were used to identify and describe the characteristics of the benthic habitats: (1) side scan sonar to identify acoustic facies, (2) Hamon grab snapshot of sediments collected in all the grab samples; (3) endofauna identified and counted after sieving the sediment on a 2 mm sieve mesh; (4) endofauna identified and counted after sieving the sediment on a 1 mm sieve mesh; (5) non-denumerable epifauna identified in grab samples; (6) sediment granulometric composition using particle size distribution; (7) species richness obtained via ROV video footage.

　　　Appendix A summarises the sampling efforts during both campaigns.

### 3.1. Side Scan Sonar Observations

　　　A complete acoustic coverage of each sampling box was carried out during the VIDEOCHARM surveys in 2010 and 2011 using side scan sonar (Figure 2) (for the methodology of the acoustic survey, see [33]). A visual analysis of the side scan sonar mosaic was conducted on board, and observations of the sediment types from grab samples and ROV footage were used to describe acoustic facies. Side scan sonar technology has been used successfully for many years to produce high-resolution acoustic maps of the seabed [17–19,33–35]. Typically, side scan sonar data are produced using a pair of transducers mounted on either side of a tow fish which is connected to a survey vessel by means of a cable. The sound emitted from the transducers ensonifies a continuous surface of seabed either side of the transducers [36]. Reflected sound received by the transducers from the surface of the seabed provides information on the nature (e.g., hardness, roughness, texture) of the sediments and the presence and disposition of seabed features (e.g., sand waves, rock outcrops, algae cover, and anthropogenic features) across the swathe [37]. For the VIDEOCHARM surveys, side scan sonar data were collected using the DF 1000 Edgetech (100–400 kHz) side scan sonar system in conjunction with data acquisition software. Data were processed, georeferenced and mosaiced using the 'Caraïbes' software package from IFREMER to produce continuous acoustic maps of the area surveyed. The vessel position was provided by a Thales 3011/Fugro SeaStar DGPS system, and the position of the side scan tow fish was calculated by using vessel heading, vessel offsets, tow cable layback and tow fish depth.

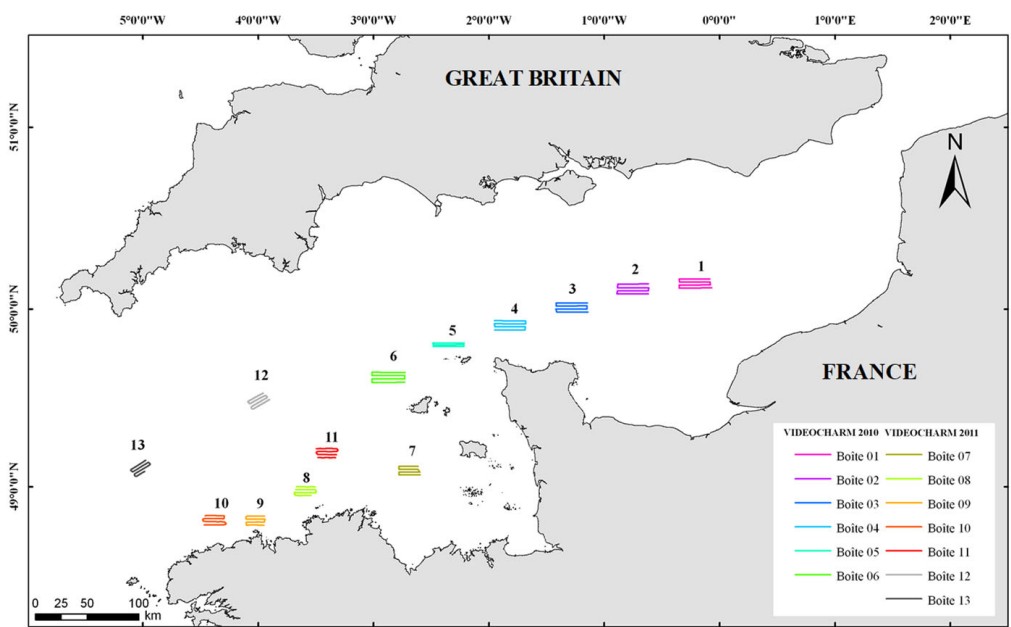

**Figure 2.** Side scan sonar profiles collected during VIDEOCHARM surveys in June 2010 and June 2011 for the 13 boxes 1 to 13.

Several morphological structures were identified from the scan sonar profiles.
- Zone with ribbons;
- Zone with furrows;
- Zone with dunes: small and medium/large and very large dunes;
- Sand veneer on rocks;
- Homogeneous zone;
- Table rocks, outcropping or sub-flush rocks;
- Rocky area;
- Presence of anthropic evidence: net traces (dredge or trawl) and wreck.

### 3.2. Grab Sample Collection

Sediment and macrofauna samples were collected with a 0.25 m$^2$ Hamon grab to ground-truth the acoustic surveys through the provision of information on sediment particle size distributions and macrofaunal communities. The grab sampling was also in the spatial continuity of the benthic sampling program CHARM II, to complement existing databases in the easternmost part of the Channel [25,38]. One replicate was used for sediment characterisation, and two replicates were used for macrofauna analyses. Snapshots of each replicate were made directly after the grabs were recovered. The two macrofauna grabs were then washed onboard the RV 'Côtes de la Manche' over two circular superposed mesh sieves (1 mm and 2 mm) to remove fine sediments. The biological and sediment content was fixed in containers with 10% buffered formaldehyde solution. Data species richness and abundances from both replicates (0.5 m$^2$) were pooled for ecological analyses, and colonial epifaunal taxa were reported only as present.

A subsample from the sediment grab was analysed for particle size distribution. First, sediment was wet sieved over a 50 μm mesh. The sieved sediment fraction (50 μm) was kept still and left to deposit for 48 h and then dried after the supernatant was removed. The rest of the sediment (>50 μm) was dried at 70 °C and then sieved using 32 sieve-column (50; 63; 80; 100; 125; 160; 200; 250; 315; 400; 500; 630; 800; 1000; 1250; 1600; 2000; 2500; 3150; 4000; 5000; 6300; 8000; 10,000; 12,500; 16,000; 20,000; 25,000; 31,500; 40,000; 50,000; 63,000 μm), and the total weight of each fraction was recorded. The sieve choice followed the modified Wentworth's classification to determine the sediment type of each station using a Folk diagram [25,26,38,39]. For each station, the sediments were characterised by

five main sedimentary fractions: pebbles >20 mm; large gravel (20–5 mm); gravel (2–5 mm), sand (2 mm–63 μm), and silt–clay (<63 μm).

### 3.3. Remotely Operated Vehicle

A small ROV Seabotix LBV 200L2 was deployed in each box, except for in Box 10. Between one and four video observations were made in each box depending on the weather conditions, allowing the ROV to operate safely. The video system included a colour camera pointing at the ground at 45° from the horizontal, with the lens about 50 cm above the seabed. A couple of laser pointers (10 cm apart) were used to estimate the size of the surface sediment and the megafauna taxa on the seabed. The duration of the video footage varied from 4 to 10 min. Some snapshots were extracted from the video and were used to identify benthic taxa. The ROV Seabotix LBV 200L2 was a small piece of equipment, which was difficult to use in a megatidal sea such as the Channel (e.g., the need to use it when the vessel was anchored in slack water), which considerably reduces the number of observations per day [12].

A total of 30 videos were recorded (the ROVbis were not analysed) (Appendix A), and it characterised the surface sediments into five classes: coarse sands, gravel, pebbles, boulders and hard bottom; and into three classes of occurrence: existing, common (present throughout the footage but sparse) and dominant (prevailing sediment class throughout the footage), according to the expertise of J.C.D.

Megafauna taxa were identified for encrusting and erect sessile fauna and the motile fauna following the expertise of J.C.D.: sponges, bryozoans, *Flustrea fasciata*, *Alcyonidium* spp., cnidarians, dead man fingers *Alcyonium digitatum,* ross coral *Pentapora fasciata*, hydroids (tuff), *Nemertesia antennina*, polychaetes, *Spirobranchus* spp., *Sabella* spp., sea urchin, *Ophiothrix fragilis*, *Asterias rubens*, decapods, gastropods, the common whelk *Buccinum undatum,* and fish. Three classes of abundance of the taxa were established: (1) rare, one to some individuals, (2) common taxa (present throughout the footage but sparse) and (3) abundant taxa (present throughout the footage in abundance).

### 3.4. Database and Statistical Analyses

Appendix A reports all the operations realised during the VIDEOCHARM surveys in 2010 and 2011. The different operations permit to obtain seven data tables, corresponding to seven habitat sampling methods: (1 and 2) endofauna collected at the 40 grab stations after sieving on 2 mm and then on 1 mm (in most cases two replicates per station, apart from four stations B21, B32, B109, B129 where only one grab was available; in these cases, the data were doubled to obtain a total surface of 0.5 m$^2$); (3) nondenumerable epifauna identified on pebbles and blocks in the 40 stations; (4) Hamon grab snapshot of sediment collected for all replicates (see Appendix A); (5) sediment particle size distribution, (6) video footage of the ROV SeaBotix at 30 stations; (7) interpretation of the side scan sonar profiles to identify acoustic facies.

Faunal data were used to calculate the taxonomic richness (TR, number of taxa per 0.5 m$^2$), abundance (number of individuals per 0.25 m$^2$), and diversity indices for each station. The Shannon–Weaver diversity index (H′) in log$_2$ and Pielou's evenness (J′) were calculated. Ecological status was estimated from diversity indices H′ and J values according to the thresholds defined previously and resumed in [39]: 0–1, bad; 1–2: poor; 2–3: moderate; 3–4: good and >4: high. For J, the thresholds are <0.2: bad; 0.2–0.4: poor; 0.4–0.6: moderate; 0.6–0.8: good and >0.8: high, which were independent from the organic matter concentration in the sediment. Data analysis was performed using the PRIMER version 6 software package (Plymouth Routines in Multivariate Ecological Research) [40].

Hierarchical cluster analysis (HCA) was carried out on the matrices based on Sorensen's coefficient for the presence/absence (all the taxa) and on Log10(x + 1) or square root transformed abundances per 0.5 m$^2$ used to down-weight the importance of very abundant denumerable taxa with the Bray–Curtis similarity using group average linking of the species found in the different stations, with the construction of dendrograms using the

group average algorithm generated from the PRIMER-6 software package (Plymouth Routines in Multivariate Ecological Research). To identify those species within different groups which primarily account for the observed assemblage differences, SIMPER (SIMilarity PERcentage) routines were performed [40].

## 4. Results

### 4.1. Side Scan Sonar Observations

Figure 2 shows the localisation of side scan sonar profiles realised during the VIDEOCHARM surveys in June 2010 and June 2011. The lengths of the profiles extend between 39.4 km in Box 7 and 84.5 km in Box 13 (Table 1). The lengths of the profiles were higher in 2010 (boxes 1 to 6) than in 2011 (boxes 7–13) due to the bad weather during the second survey.

Figure 3 gives examples of the different structures observed along the side sonar profiles in three selected boxes along the east (Box 1), central (Box 3) to west (Box 13) gradient. The central part of EC North Cotentin peninsula is known to be an area of high hydrodynamic. Sediment coverage and bottom-recognized structures in this part of the EC highlight high gradients of variability.

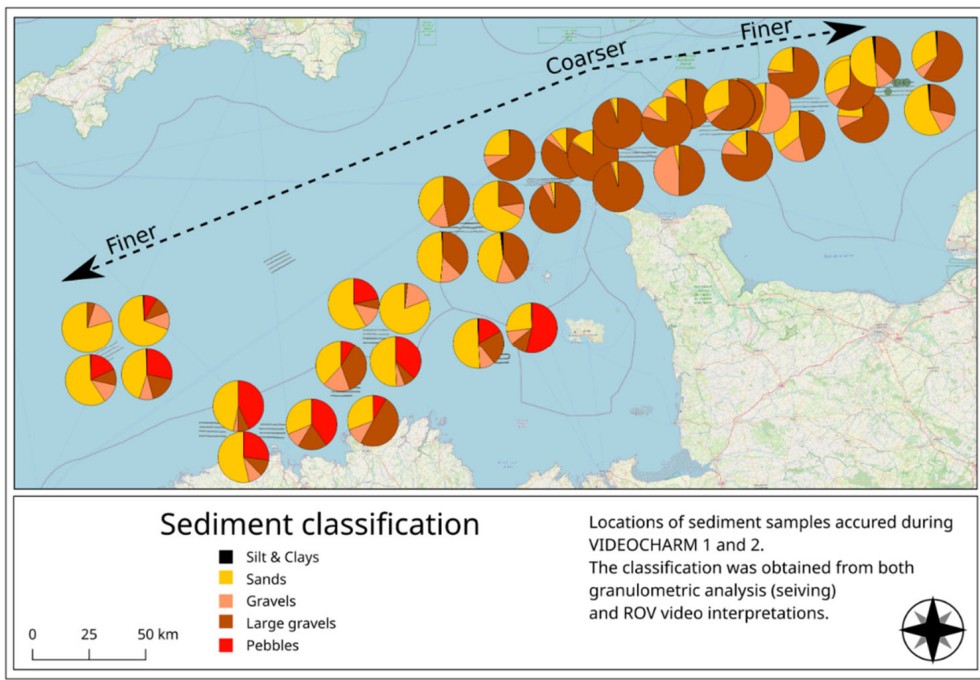

**Figure 3.** Five main sediment classes described during the VIDEOCHARM surveys in June 2010 and June 2011.

Within Box 1, side scan sonar profiles showed a relative homogeneity of the acoustic reflectivity. However, two different acoustic responses could be identified (Figure 3): rugged acoustic facies to the west and a smoother acoustic aspect to the east. Snapshot pictures extracted from ROV footage highlighted a relative homogeneity of the sediment cover with gravel mainly present. Coarse sands and pebbles were also present, acting as an armoured structure over a trapped coarse sand content. This coarsening gradient from west to east is likely to explain the different acoustic returns, i.e., the coarser the cover, the more homogeneous the acoustic response.

Sonograms are more contrasted in Box 3, located to the northeast of the Cotentin peninsula (Figure 3). Surface sediments were mostly coarse gravel and pebbles. Some longitudinal sedimentary features can be seen to the southeast of the box area. Their orientation, parallel to the tide, indicates a high current velocity allowing for movements of coarse sediments (mixed coarse sands and gravel). Additionally, some rocky outcrops

can be identified to the south and east of Box 3, as well as in the centre. The movements of biogenic coarse sands, again under the effects of high current velocities, could recover and mask rocky outcrops explaining the intermittent appearance of these outcrops as the sedimentary cover is very thin in the area (i.e., a few centimetres).

Box 13 (around 100 m depth) is located offshore the Brittany coast, at the Channel entry, in deeper waters (~100 m). Some sedimentary features showed dunes and mega-dunes formed with coarse sands and gravel (Figure 3). At these locations, the ROV survey showed a succession of dunes of about 1.5–2 m high and 50–60 m wavelength. Some sedimentary figures were organised in crescent shape, showing both a high level of hydro-sedimentary dynamic and the direction of transport (towards the inner parts of the EC).

*4.2. Sediment*

Particle size analyses showed that all the sampling stations were classed as coarse sediment, ranging from sandy gravel to pebbles (Appendix B; Figure 4). The percentage of fine particles (<63 µm) was very low and varied from 0% to 1.89% of the dry sediment, with the highest percentage recorded in samples classed as sandy gravel. The percentage of sand varied strongly (3.5% to 80.9%), similarly to the percentage of gravel (2.6% to 54.7%) and large gravel (0.3% to 94.5%) (Appendix B). The pebbles were present only in the westernmost boxes offshore the Brittany coast (8.4% to 53.6%).

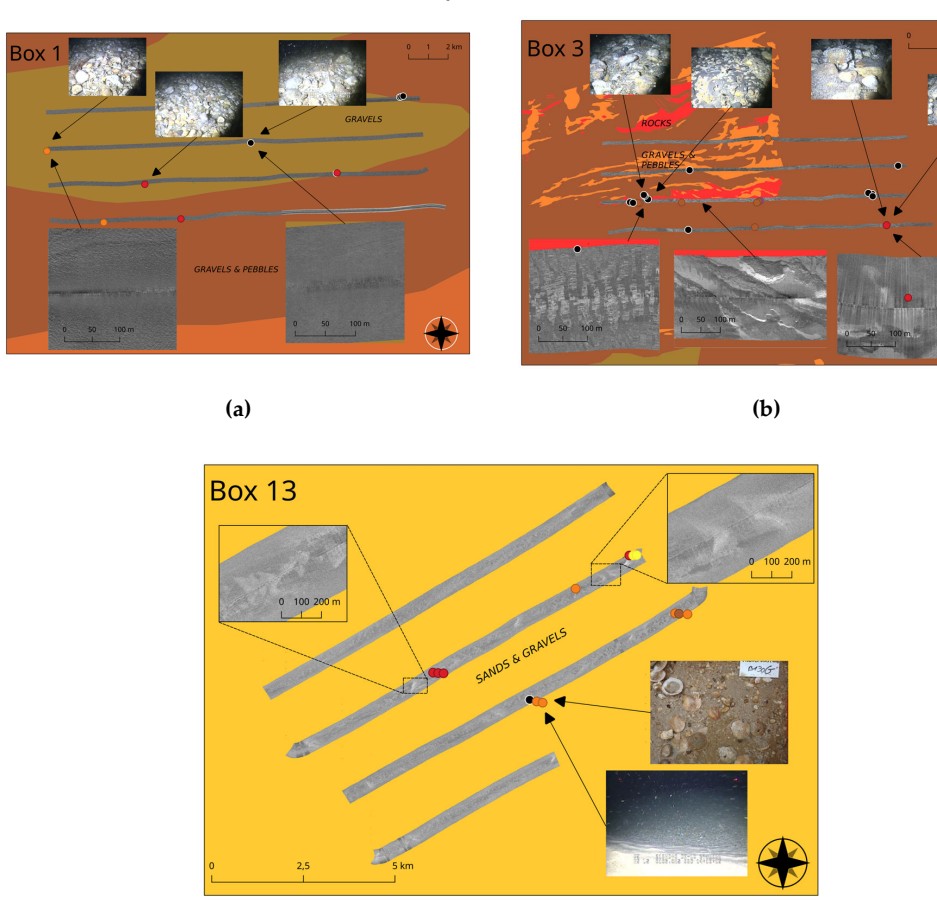

**Figure 4.** Example of scan sonar profiles collected during the VIDEOCHARM surveys in June 2010 ((**a**,**b**) boxes 1 and 3) and June 2011 ((**c**) box 13) with inset views for some areas of interest and snapshots of the surface sediments extracted from the ROV video footage.

Sediments were classified into four main types (Table 2): gravel (11 stations located in boxes 3, 4 and 5), gravelly sand (two stations located in boxes 11 and 13); sandy gravel (15 stations located in boxes 1, 2, 3, 5 and 6) and sandy gravel and pebbles (12 stations

located in boxes 7, 8, 9, 10, 11 and 13). Two sedimentary gradients were observed from each end of the transect to the central part of the EC, with sandy gravel and pebbles in the western part of the EC, gravel in the middle of the EC, and sandy gravel in the eastern part of the EC.

### 4.3. General Patterns of the Fauna

The total number of taxa including countable barnacles was 272. Among them, 233 taxa were recorded only on the 2 mm mesh size, 138 on 1 mm and 95 were recorded both on 2 and 1mm mesh sieve mesh. A total of 138 taxa were recorded only on 2 mm mesh size, and only 43 were recorded on 1 mm. The total richness included 110 Polychaeta (40% of the total taxa), 90 Crustacea (33%), 49 Mollusca (18%), 11 Echinoderms (4%) and 12 others (5%). The total number of individuals was 11,626 individuals, with 7425 on 2 mm (63%) and 5201 on 1 mm (37%). The dominant species was the barnacle *Balanus crenatus* (1644 individuals, 14% of the individuals), and the 10 more abundant taxa represented 40% of the fauna and the 20 more abundant taxa 60%. Among these 20-dominant taxa, 12 were Polychaeta, three Crustacea, three Bivalves and two Echinoderms. An additional 125 colonial taxa were identified in the epifauna found in the grab Hamon grab samples, with a dominance of Bryozoans and Hydrozoans.

A total of 34 invertebrate taxa were identified on the snapshots taken from the ROV footage, and amongst them, 12 taxa were not recorded in grab samples such as the Bryozoans *Alcyonidium* spp., the Cnidaria *Alcyonium digitatum*, and the large echinoderms *Asterias rubens*, *Echinocardium pennatifidum*, *Echinus esculentus*, *Henricia sanguinea*, *Ophiocomina nigra* and *Sollaster paposus*.

Overall, a total of 409 taxa was recorded during this study on the coarse sediment from the deeper part of the English Channel.

Univariate indices (TR per 0.5 m$^2$, abundance per 0.25 m$^2$, H' and J) for the accounted taxa recorded on 2 mm and 1 + 2 mm sieve meshes are shown in Table 2. On 2 mm, the TR varied from two taxa at station 31 (Box 5) to 73 taxa at station 5 (Box 1), whilst on 1 mm the TR varied from two taxa at station 31, which was the poorest to 88 at station 28 (Box 6), which was the most diversified (Table 2). The number of taxa recorded per station increased from 0 to 34 when the sieving mesh was reduced to 1 mm, and the mean number of taxa per 0.5 m$^2$ was 36.35 on 2 mm to 51.08 on 1 mm, i.e., a 40% increase. On 1 mm, the mean TR per box varied from 24.5 for 0.5 m$^2$ in Box 5 to 76 for 0.5 m$^2$ in boxes 1 and 9; the poorest TR per box for 0.5 m$^2$ varied from 24.5 to 34 in boxes 5, 7 and 8, while the richest TR per box for 0.5 m$^2$ varied from 62 to 76 in boxes 1, 6 and 9. No west–east gradient was observed.

On 2 mm, the mean abundance per 0.25 m$^2$ varied from 11 at station 23 (Box 6) to 392.5 at station 5 (Box 5) both on sandy gravel, while on 1 mm, it varied from 20 at station 108 (Box 8) to 445 at station 5 (Table 2). The mean abundance per 0.25 m$^2$ on 2 mm was 93.31 and 145.33 on 1 mm, i.e., an increase of 52%. On 1 mm, the mean abundances per 0.25 m$^2$ varied from 36 in Box 8 to 261 in Box 1. The two eastern boxes 1 and 2 exhibited a mean abundance higher than 200 ind. 0.25 m$^2$. Five boxes (3, 4, 5, 6 and 9) showed a mean abundance included between 100, while the five last boxes showed a mean abundance lower than 100 ind. 0.25 m$^2$: boxes 7, 8, 10, 11 and 13.

On 2 mm, the Pielou's evenness J' varied from 0.36 to 0.95 corresponding to an ecological status ranging from poor to high. Of the 40 stations, 18 were classified as high, 14 as good, six as moderate and two as poor (Table 2). On 1 mm, J' varied from 0.40 to 0.96; 25 stations were classified as high, 13 as good and two as moderate. On 2 mm, the Shannon–Weaver diversity index (H') varied from 0.26 to 2.73 corresponding to an ecological status ranging from bad to moderate. Of the 40 stations, 18 were classified as moderate, 19 as poor and 3 as bad. On 1 mm, H' varied from 0.4 to 5.12; 21 stations were classified as high, 13 as good, four as moderate, one as poor and one as bad (Table 2).

**Table 2.** Univariate indices value and ecological quality status for the 40 stations classified per main sediment type and boxes according to the taxa accounted for on 2 mm and 1 + 2 mm mesh sieves. TR (taxonomic richness), total number of species recorded on 0.5 m$^2$; A: total abundance per 0.25 m$^2$; J': Pielou's evenness and H': Shannon–Weaver diversity. The colour coding corresponds to the Ecological Status of the Water Framework Directive: blue, high status; green, good status; yellow, moderate status; orange, poor status, and red bad status.

| Main Sediment Type | Box | Station | 2 mm | | | | 2 + 1 mm | | | |
|---|---|---|---|---|---|---|---|---|---|---|
| | | | TR | A | J' | H' | TR | A | J' | H' |
| Gravel | 3 | 11 | 53 | 104.0 ± 25.5 | 0.74 ± 0.14 | 2.17 ± 0.82 | 75 | 234.0 ± 25.5 | 0.85 ± 0.03 | 4.97 ± 0.44 |
| | | 12 | 60 | 135.0 ± 36.8 | 0.80 ± 0.02 | 2.49 ± 0.03 | 74 | 265.0 ± 36.8 | 0.84 ± 0.02 | 4.93 ± 0.11 |
| | | 14 | 54 | 184.5 ± 51.6 | 0.65 ± 0.01 | 2.10 ± 0.10 | 76 | 286.5 ± 88.4 | 0.77 ± 0.01 | 4.42 ± 0.15 |
| | | 15 | 21 | 19.0 ± 21.2 | 0.95 ± 0.01 | 1.41 ± 1.12 | 37 | 44.0 ± 36.8 | 0.94 ± 0.01 | 4.08 ± 0.82 |
| | 4 | 18 | 64 | 132.5 ± 40.3 | 0.83 ± 0.01 | 2.70 ± 0.17 | 87 | 195.5 ± 27.6 | 0.86 ± 0.01 | 5.02 ± 0.11 |
| | | 19 | 47 | 282.0 ± 0.0 | 0.66 ± 0.00 | 2.31 ± 0.00 | 81 | 423.0 ± 164.0 | 0.74 ± 0.07 | 4.46 ± 0.69 |
| | | 20 | 14 | 11.5 ± 3.5 | 0.97 ± 0.01 | 1.51 ± 0.23 | 27 | 39.5 ± 3.5 | 0.93 ± 0.01 | 4.24 ± 0.17 |
| | | 21 | 27 | 52.0 ± 12.7 | 0.74 ± 0.03 | 1.70 ± 0.06 | 38 | 93.0 ± 12.7 | 0.82 ± 0.03 | 3.98 ± 0.18 |
| | 5 | 31 | 2 | 150.0 ± 0.0 | 0.40 ± 0.00 | 0.26 ± 0.00 | 2 | 150.0 ± 0.0 | 0.40 ± 0.00 | 0.40 ± 0.00 |
| | | 36 | 29 | 37.0 ± 1.4 | 0.87 ± 0.04 | 2.01 ± 0.05 | 55 | 83.0 ± 29.7 | 0.89 ± 0.01 | 4.58 ± 0.29 |
| | | 38 | 22 | 65.5 ± 20.5 | 0.45 ± 0.63 | 1.14 ± 1.62 | 34 | 112.5 ± 36.1 | 0.72 ± 0.22 | 3.27 ± 1.27 |
| Gravelly Sand | 11 | 126 | 8 | 15.5 ± 2.1 | 0.66 ± 0.09 | 0.77 ± 0.38 | 12 | 24.0 ± 0.0 | 0.83 ± 0.01 | 2.49 ± 0.48 |
| | 13 | 131 | 14 | 28.5 ± 30.4 | 0.81 ± 0.27 | 1.21 ± 0.01 | 26 | 53.5 ± 53.0 | 0.81 ± 0.24 | 3.18 ± 0.63 |
| Sandy Gravel | 1 | 2 | 43 | 175.0 ± 147.1 | 0.60 ± 0.46 | 1.74 ± 1.41 | 69 | 230.0 ± 144.2 | 0.66 ± 0.34 | 3.69 ± 2.08 |
| | | 3 | 52 | 103.0 ± 17.0 | 0.89 ± 0.01 | 2.73 ± 0.02 | 68 | 170.0 ± 22.6 | 0.85 ± 0.01 | 4.80 ± 0.19 |
| | | 4 | 63 | 123.5 ± 81.3 | 0.83 ± 0.17 | 2.63 ± 0.33 | 83 | 198.5 ± 94.0 | 0.86 ± 0.10 | 5.17 ± 0.52 |
| | | 5 | 73 | 392.5 ± 200.1 | 0.66 ± 0.17 | 2.39 ± 0.51 | 81 | 445.0 ± 250.3 | 0.69 ± 0.13 | 4.06 ± 0.58 |
| | 2 | 6 | 39 | 161.0 ± 161.2 | 0.56 ± 0.14 | 1.47 ± 0.06 | 46 | 173.0 ± 161.2 | 0.61 ± 0.18 | 2.99 ± 0.70 |
| | | 7 | 44 | 147.5 ± 74.2 | 0.73 ± 0.03 | 2.21 ± 0.16 | 58 | 239.5 ± 74.2 | 0.79 ± 0.03 | 4.41 ± 0.03 |
| | | 8 | 27 | 119.0 ± 45.3 | 0.36 ± 0.10 | 0.84 ± 0.51 | 45 | 148.5 ± 54.4 | 0.53 ± 0.06 | 2.56 ± 0.61 |
| | | 9 | 47 | 80.5 ± 3.6 | 0.85 ± 0.01 | 2.43 ± 0.02 | 79 | 262.5 ± 17.7 | 0.81 ± 0.05 | 4.66 ± 0.31 |
| | 3 | 10 | 7 | 39.0 ± 5.7 | 0.76 ± 0.01 | 1.05 ± 0.15 | 10 | 56.0 ± 7.1 | 0.79 ± 0.01 | 2.33 ± 0.45 |
| | | 16 | 46 | 68.5 ± 14.8 | 0.90 ± 0.01 | 2.58 ± 0.10 | 63 | 140.0 ± 46.7 | 0.88 ± 0.06 | 4.83 ± 0.13 |
| | 6 | 22 | 66 | 145.0 ± 76.4 | 0.68 ± 0.01 | 2.18 ± 0.33 | 76 | 156.5 ± 74.2 | 0.71 ± 0.01 | 3.95 ± 0.23 |
| | | 23 | 14 | 11.0 ± 1.4 | 0.95 ± 0.01 | 1.39 ± 0.09 | 26 | 22.5 ± 0.7 | 0.96 ± 0.01 | 3.94 ± 0.17 |
| | | 24 | 43 | 44.0 ± 35.3 | 0.93 ± 0.02 | 2.26 ± 0.66 | 58 | 89.0 ± 50.9 | 0.89 ± 0.04 | 4.56 ± 0.39 |
| | | 28 | 56 | 94.5 ± 47.4 | 0.90 ± 0.05 | 2.76 ± 0.15 | 88 | 278.5 ± 88.4 | 0.82 ± 0.02 | 4.84 ± 0.01 |
| | 5 | 32 | 7 | 67.0 ± 0.0 | 0.64 ± 0.00 | 1.12 ± 0.00 | 7 | 67.0 ± 0.0 | 0.64 ± 0.00 | 1.80 ± 0.00 |
| Sandy Gravel and Pebbles | 7 | 102 | 26 | 61.5 ± 3.5 | 0.68 ± 0.04 | 1.62 ± 0.10 | 35 | 79.0 ± 14.1 | 0.76 ± 0.01 | 3.49 ± 0.04 |
| | | 105 | 19 | 70.5 ± 9.2 | 0.52 ± 0.07 | 1.10 ± 0.22 | 33 | 105.5 ± 19.1 | 0.68 ± 0.15 | 3.20 ± 1.04 |
| | 8 | 108 | 9 | 16.0 ± 0.0 | 0.90 ± 0.00 | 1.46 ± 0.00 | 13 | 20.0 ± 0.0 | 0.92 ± 0.00 | 3.40 ± 0.00 |
| | | 109 | 25 | 33.5 ± 34.6 | 0.89 ± 0.07 | 1.68 ± 0.64 | 35 | 52.0 ± 52.3 | 0.91 ± 0.07 | 3.76 ± 0.62 |
| | 9 | 114 | 59 | 72.5 ± 21.9 | 0.92 ± 0.05 | 2.68 ± 0.17 | 76 | 154.5 ± 21.9 | 0.88 ± 0.03 | 5.12 ± 0.10 |
| | 10 | 119 | 35 | 45.5 ± 17.7 | 0.86 ± 0.06 | 2.08 ± 0.09 | 47 | 74.0 ± 53.7 | 0.90 ± 0.01 | 4.27 ± 0.65 |
| | | 120 | 41 | 58.5 ± 82.7 | 0.44 ± 0.62 | 1.42 ± 2.01 | 68 | 99.5 ± 88.4 | 0.91 ± 0.04 | 4.65 ± 0.73 |
| | 11 | 125 | 35 | 73.0 ± 14.1 | 0.71 ± 0.14 | 1.79 ± 0.41 | 42 | 92.0 ± 8.5 | 0.75 ± 0.13 | 3.48 ± 0.83 |
| | | 129 | 32 | 41.0 ± 18.4 | 0.78 ± 0.22 | 1.77 ± 0.47 | 37 | 72.0 ± 18.4 | 0.84 ± 0.09 | 3.99 ± 0.39 |
| | | 130 | 57 | 156.0 ± 219.2 | 0.39 ± 0.55 | 1.44 ± 2.03 | 68 | 192.0 ± 236.2 | 0.84 ± 0.11 | 4.06 ± 0.78 |
| | 13 | 133 | 29 | 43.5 ± 44.6 | 0.84 ± 0.10 | 1.74 ± 1.19 | 40 | 69.5 ± 44.5 | 0.82 ± 0.12 | 3.64 ± 1.53 |
| | | 134 | 45 | 73.0 ± 2.8 | 0.89 ± 0.03 | 2.58 ± 0.06 | 68 | 123.0 ± 29.7 | 0.90 ± 0.01 | 5.01 ± 0.25 |

### *4.4. Pattern of the 2 mm Macrofauna*

The Hierarchical Cluster Analysis CA (not shown in this paper) within the Sorensen coefficient (presence/absence of the 239 taxa in the 20 stations) identified the presence of

16 groups of stations at 40% similarity without clear sedimentological and geographical patterns. The HCA within the Log(X + 1) transformation of the abundances identified seven groups of stations (24% of the Bray–Curtis similarity) (Figure 5). Two of the groups included a large number of stations characterised by high diversity, whilst the five other groups were characterised by a fewer number of species locally present in high abundance. The faunal group a included two stations from Box 5, characterised by two taxa, the bivalve *Pododesmus squama* and the barnacle *Balanus crenatus* (Table 3). Group b included only station 10, Box 3. Group c included two stations of the western boxes 11 and 13 characterised by two echinoderms *Echinocyamus pusillus* and *Spatangus purpureus* (Table 3). Group d corresponded to the isolated station 23 (Box 6) characterised by the bivalve *Glycymeris glycymeris*. Group e regrouped four stations from boxes 4 and 7 with abundant populations of the brittle star *Ophiothrix fragilis*. Group f included 10 stations from western boxes 8–13, characterised by the sea urchin *Echinocyamus pusillus*, the polychaetes *Glycera lapidum* and *Jasmineira elegans* and the amphipod *Ampelisca spinipes*. Finally, group g included 20 stations (50% of the stations) mainly from boxes 1 and 2 in the eastern part and stations from boxes 4, 5 and 6 in the central part of the English Channel. They were stations with high taxonomic richness; the SIMPER analysis showed that polychaetes *Glycera lapidum, Notomastus latericeus* and *Lumbrineris gracilis* and the barnacle *Balanus crenatus* were the main species contributing to this group.

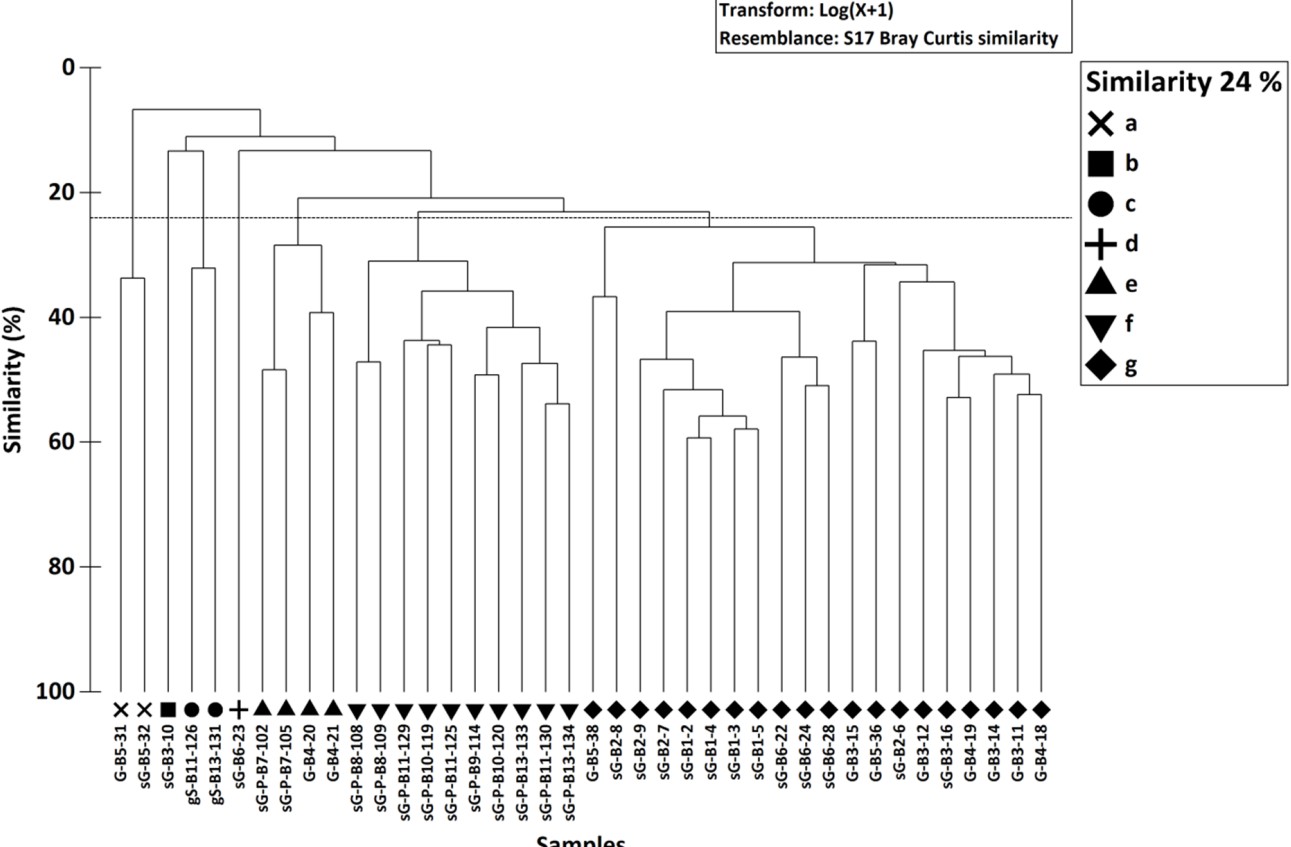

**Figure 5.** Cluster dendrogram showing the pattern of the 40 grab sampling stations (abundance per 0.5 m$^2$ of the accounted macrofauna retained on a 2 mm mesh sieve and 24% of similarity) according to the Bray–Curtis similarity after Log(X + 1) transformation of the abundances.

**Table 3.** SIMPER analysis on taxa accounted for on 2 mm mesh sieve with cumulative contribution (Cc in%) of the ten top species in the different groups identified by the cluster dendrogram analysis (Figure 5). (1): Group 1, ... of Table 7. Group b (station 10, group 2). Group d (station 23, group 6).

| Group a (1) | Cc (%) | Group c (2) | Cc (%) | Group e (5) | Cc (%) | Group f (4) | Cc (%) | Group g (3) | Cc (%) |
|---|---|---|---|---|---|---|---|---|---|
| *Pododesmus squama* | 61.44 | *Echinocyamus pusillus* | 64.91 | *Ophiothrix fragilis* | 26.90 | *Echinocyamus pusillus* | 17.68 | *Notomastus latericeus* | 8.84 |
| *Balanus crenatus* | 100 | *Schistomeringos neglecta* | 82.46 | *Aonides paucibranchiata* | 40.31 | *Glycera lapidum* | 26.16 | *Balanus crenatus* | 16.59 |
| | | *Spatangus purpureus* | 100 | *Laonice bahusiensis* | 52.70 | *Jasmineira elegans* | 31.76 | *Lumbrineris gracilis* | 21.71 |
| | | | | *Notomastus latericeus* | 60.64 | *Ampelisca spinipes* | 36.34 | Nemertea | 25.84 |
| | | | | *Lumbrineris gracilis* | 67.94 | *Eunice vittata* | 40.76 | *Glycera lapidum* | 29.85 |
| | | | | *Aonides oxycephala* | 73.19 | *Laonice bahusiensis* | 45.02 | Syllis spp. | 33.27 |
| | | | | *Eualus occultus* | 78.16 | *Polycirrus medusa* | 49.07 | *Laonice bahusiensis* | 36.37 |
| | | | | *Pisidia longicornis* | 80.72 | *Cheirocratus intermedius* | 53.07 | *Aonides paucibranchiata* | 39.20 |
| | | | | *Glycymeris glycymeris* | 83.22 | *Timoclea ovata* | 56.99 | *Glycymeris glycymeris* | 41.95 |
| | | | | *Timoclea ovata* | 85.73 | *Pagurus cuanensis* | 60.56 | *Pisidia longicornis* | 44.63 |

### 4.5. Pattern of the 1 + 2 mm Macrofauna

The HCA using the Sorensen coefficient (not shown in the paper) on the matrix of 40 stations with 273 taxa identified 10 groups of stations at a 40% similarity without clear sedimentological and geographical patterns. The HAC within the Log(X + 1) transformation of the abundances identified five groups of stations at a level of 29% of the Bray–Curtis similarity (Figure 6). Group a from our study was similar to group a of the previous analysis gathering two stations from Box 5 characterised by two taxa the bivalve *Pododesmus squama* and the barnacle *Balanus crenatus* (Table 4). Group b gathered 24 stations from the eastern and central boxes 1 to 7 characterised by the polychaetes *Notomastus latericeus*, *Glycera lapidum*, *Aonides paucibranchiata* and *Eulalia mustela* (Table 4), comparable to group g in the previous analysis. Group c included 11 stations from western boxes 8 to 13 and was characterised by the sea urchin *Echinocyamus pusillus*, the polychaetes *Glycera lapidum* and *Eulalia mustela* and the amphipod *Ampelisca spinipes*, comparable to group f in the previous analysis. Group d was comparable to the isolated station 23 from Box 6 (which was also isolated in the previous analysis). The last group e included both stations 10 and 126 which were also separated from all stations in the previous analysis. This last group was characterised by the small annelids *Polygordius* and *Pisione remota*.

### 4.6. Pattern of the 1 + 2 mm Macrofauna + Colonial Epifauna Taxa

A total of 393 taxa were considered for the construction of the HCA using the Sorensen coefficient (presence/absence of the taxa) revealing the presence of five groups of stations at 35% similarity (Figure 7). Group a included stations 31 and 32 (Box 5) which were separated into different groups in the previous analyses. Both stations were characterised by two taxa: the bivalve *Pododesmus squama* and the barnacle *Balanus crenatus* (Table 5). Group b included three stations, 10 and 126 which were separated from other stations in both previous analyses, plus station 108 (Box 8). This group was characterised by the small annelids *Glycera lapidum*, *Polygordius* spp., *Pisione remota* and *Syllis* spp., the bivalve *Timoclea ovata* and the sea urchin *Echinocyamus pusillus*. Group c included 18 stations from boxes 1 to 4 corresponding to groups g and b from the previous analyses. This group was characterised only by polychaetes such as *G. lapidum*, *Notomastus latericeus* and *Laonice bahusiensis* and Nemertea (Table 5). Group d included 10 of the westernmost stations, mainly from boxes 9 to 13 characterised by *E. pusillus*, small polychaetes taxa such as *Eulalia mustela*, *G. lapidum*, *Caulleriella alata* and the amphipods *Cheirocratus* ssp. and *Ampelisca*

*spinipes*. Lastly, group e included seven stations from boxes 3 to 8, again characterised by small polychaetes such as *E. mustela*, *G. lapidum* and *N. latericeus,* Nemertea, and the bivalve *Glycymeris glycymeris* as well as the brittle star *Ophiothrix fragilis* (Table 5).

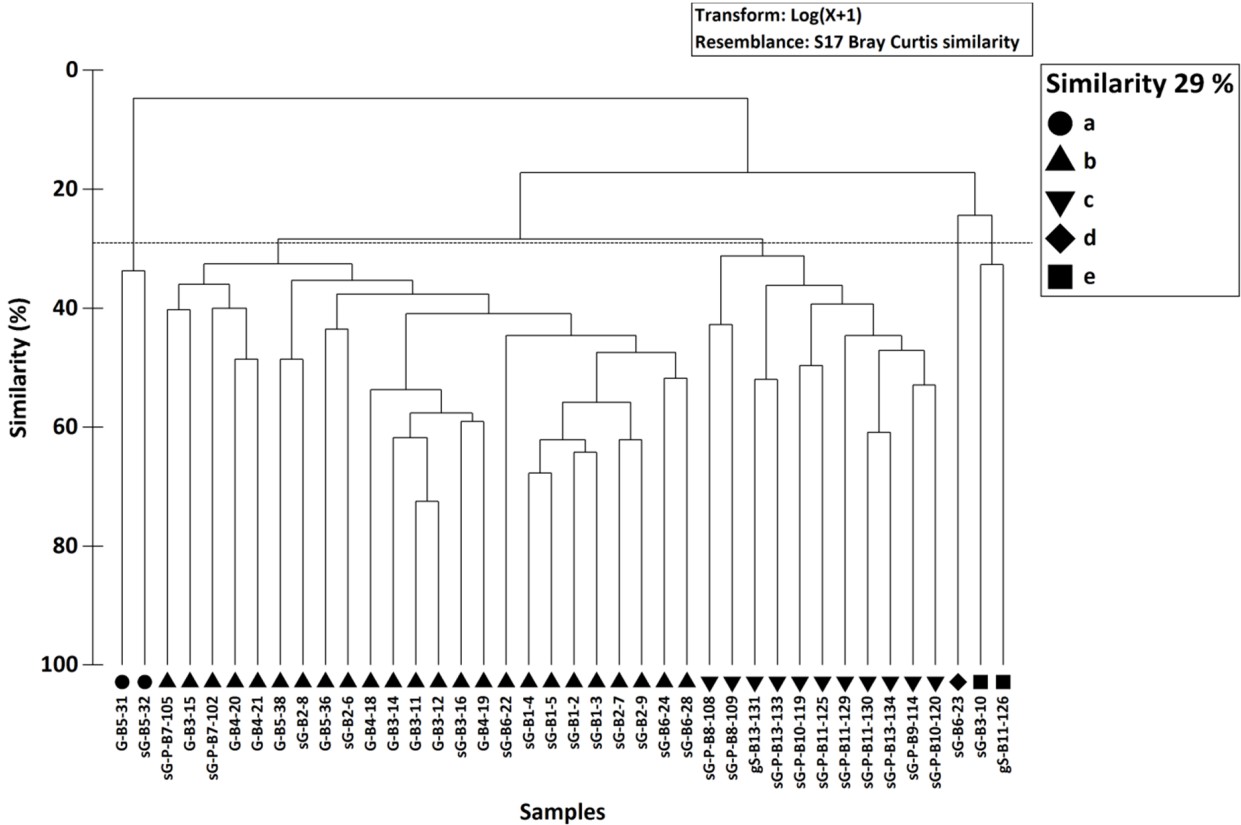

**Figure 6.** Cluster dendrogram showing the pattern of the 40 grab sampling stations (abundance per 0.5 m² of the accounted macrofauna retained on a 1 mm mesh sieve and 29% of similarity) according to the Bray–Curtis similarity after Log(X + 1) transformation of the abundances.

**Table 4.** SIMPER analysis on taxa accounted for on 2 + 1 mm mesh sieve with cumulative contribution (Cc in%) of the ten top species in the different groups identified by the cluster dendrogram analysis (Figure 6). (1): Group 1, ... of Table 7. Group b (station 10, group 2). Group d (station 23, group 6).

| Group a (1) | Cc (%) | Group b (3) | Cc (%) | Group c (4) | Cc (%) | Group e (2) | Cc (%) |
|---|---|---|---|---|---|---|---|
| *Pododesmus squama* | 61.44 | *Notomastus latericeus* | 6.62 | *Echinocyamus pusillus* | 12.48 | *Polygordius* | 35.19 |
| *Balanus crenatus* | 100.00 | *Glycera lapidum* | 11.78 | *Glycera lapidum* | 21.28 | *Pisione remota* | 55.06 |
| | | *Aonides pau-cibranchiata* | 16.56 | *Ampelisca spinipes* | 26.00 | *Syllis* spp. | 74.93 |
| | | *Eulalia mustela* | 21.22 | *Eulalia mustela* | 30.38 | *Glycera lapidum* | 87.46 |
| | | *Syllis* spp. | 25.75 | *Cheirocratus* | 34.43 | *Timoclea ovata* | 100 |
| | | *Lumbrineris gracilis* | 29.69 | *Jasmineira elegans* | 38.27 | - | - |
| | | *Laonice bahusiensis* | 33.54 | *Syllis* spp. | 41.95 | - | - |
| | | *Timoclea ovata* | 36.87 | *Laonice bahusiensis* | 44.93 | - | - |
| | | Nemertea | 40.08 | *Timoclea ovata* | 47.59 | - | - |
| | | *Balanus crenatus* | 43.09 | *Eurydice pulchra* | 50.24 | - | - |

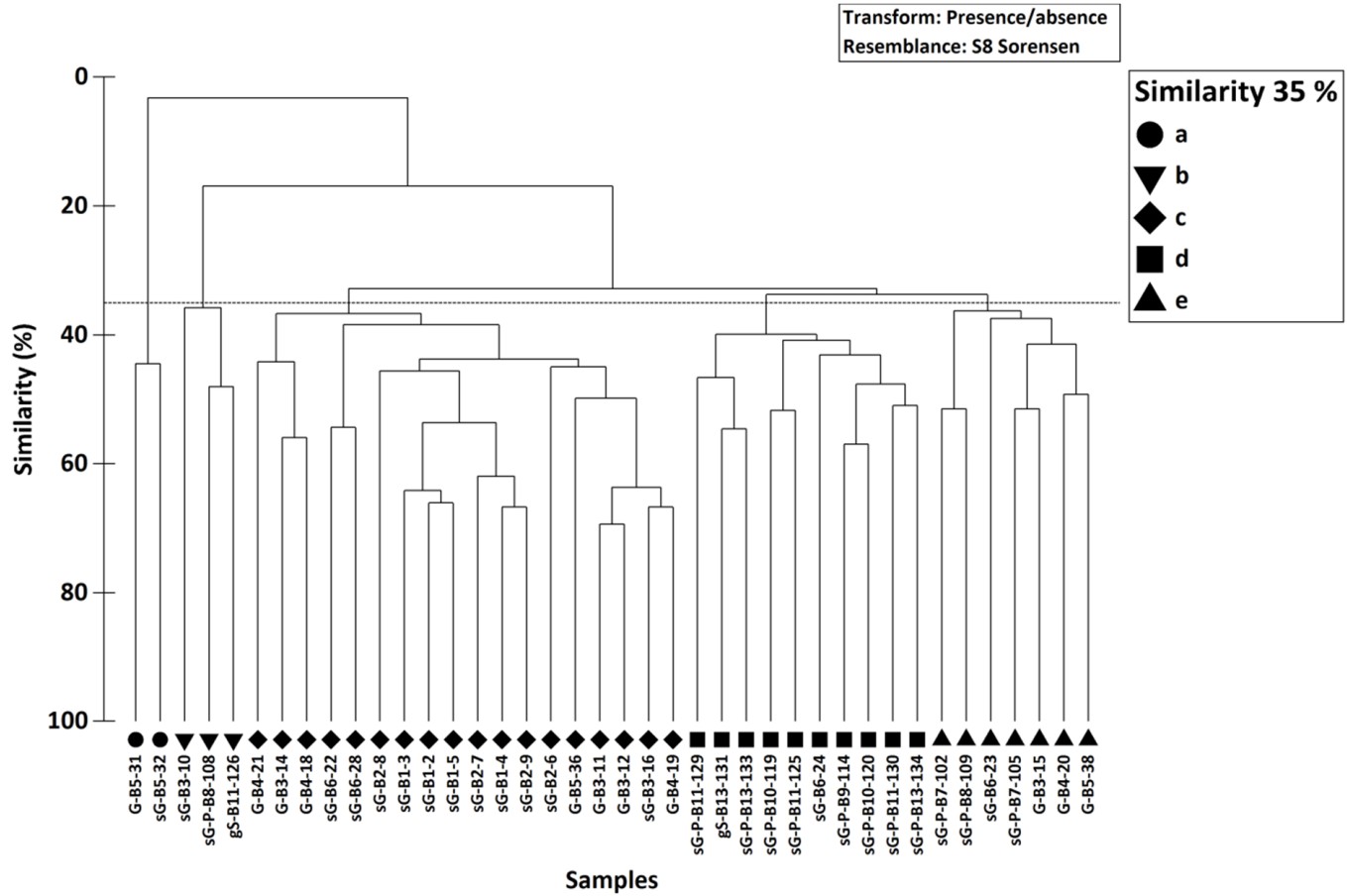

**Figure 7.** Cluster dendrogram showing the pattern of the 40 grab sampling stations (presence/absence of all taxa recorded per 0.5 m² of the motile and sessile macrofauna and 35% of similarity) according to the Sorensen similarity.

**Table 5.** SIMPER analysis on all sessile and motile taxa accounted on 2 + 1 mm mesh sieve with cumulative contribution (Cc in%) of the ten top species in the different groups identified by the cluster dendrogram analysis (Figure 7). (1): Group 1, . . . of Table 7.

| Group a (1) | Cc (%) | Group b (2) | Cc (%) | Group c (3) | Cc (%) | Group d (4) | Cc (%) | Group e (5) | Cc (%) |
|---|---|---|---|---|---|---|---|---|---|
| *Balanus crenatus* | 50.0 | *Glycera lapidum* | 21.6 | *Glycera lapidum* | 2.8 | *Echinocyamus pusillus* | 4.5 | *Eulalia mustela* | 7.8 |
| *Pododesmus squama* | 100.0 | *Polygordius* sp. | 43.1 | *Notomastus latericeus* | 5.7 | *Eulalia mustela* | 9.0 | *Glycera lapidum* | 15.7 |
| - | - | *Timoclea ovata* | 64.7 | *Laonice bahusiensis* | 8.3 | *Glycera lapidum* | 13.5 | Nemertea | 23.6 |
| - | - | *Pisione remota* | 72.3 | Nemertea | 10.8 | *Cheirocratus* | 17.1 | *Notomastus latericeus* | 31.4 |
| - | - | *Syllis* spp. | 79.9 | *Syllis* spp. | 13.3 | *Caulleriella alata* | 20.6 | *Aonides paucibranchiata* | 37.1 |
| - | - | *Echinocyamus pusillus* | 86.6 | *Eulalia mustela* | 15.7 | *Ampelisca spinipes* | 24.1 | *Syllis* spp. | 42.8 |
| - | - | *Malmgreniella arenicolae* | 93.3 | *Lumbrineris gracilis* | 18.2 | *Eunice vittata* | 27.5 | Opisthodonta | 48.4 |
| - | - | - | - | *Polycirrus medusa* | 20.6 | *Laonice bahusiensis* | 30.8 | *Polygordius* sp. | 52.3 |
| - | - | - | - | *Aonides paucibranchiata* | 22.9 | Nemertea | 34.1 | *Glycymeris glycymeris* | 56.0 |
| - | - | - | - | *Websterinereis glauca* | 25.1 | *Syllis* spp. | 37.5 | *Ophiothrix fragilis* | 59.7 |

### 4.7. ROV Observation

A total of 35 invertebrate taxa were identified from the 30 ROV videos. The Bray–Curtis similarity was calculated using the three classes of abundance ((1) rare; (2) common; (3) abundant). The HCA within the square root transformation permitted to identify six groups of stations at a 39% level of similarity (Figure 8). Station ROV111 (Group a) with only two identified taxa (*Spirobranchus* spp. and *Pisa* spp.) was isolated from the other stations. Two stations ROV13 and ROV112 (group b) were characterised by the Bryozoan *Alcyonidium* sp. (Table 6). Group c also included two stations ROV9 and ROV103, with dense populations of the brittle star *Ophiothrix fragilis* accompanied by *Alcyonium digitatum*, Hydroids and *Urticana felina* (Table 6). Group d gathered eight stations mainly located in the eastern boxes and characterised by the queen scallop *Aequipecten opercularis*. Group e included five stations all located in the western part of the English Channel and characterised by the Hydrozoa *Abietinaria abietina*. Finally, group f included 12 stations both in the central boxes and the western boxes and characterised by a diversified epifauna and the dominance of the Bryozoan *Flustra* and sponge taxa. The arrangement of the stations appeared to be governed primarily by the faunal composition and the geographical location rather than the sediment characteristics (Figure 7).

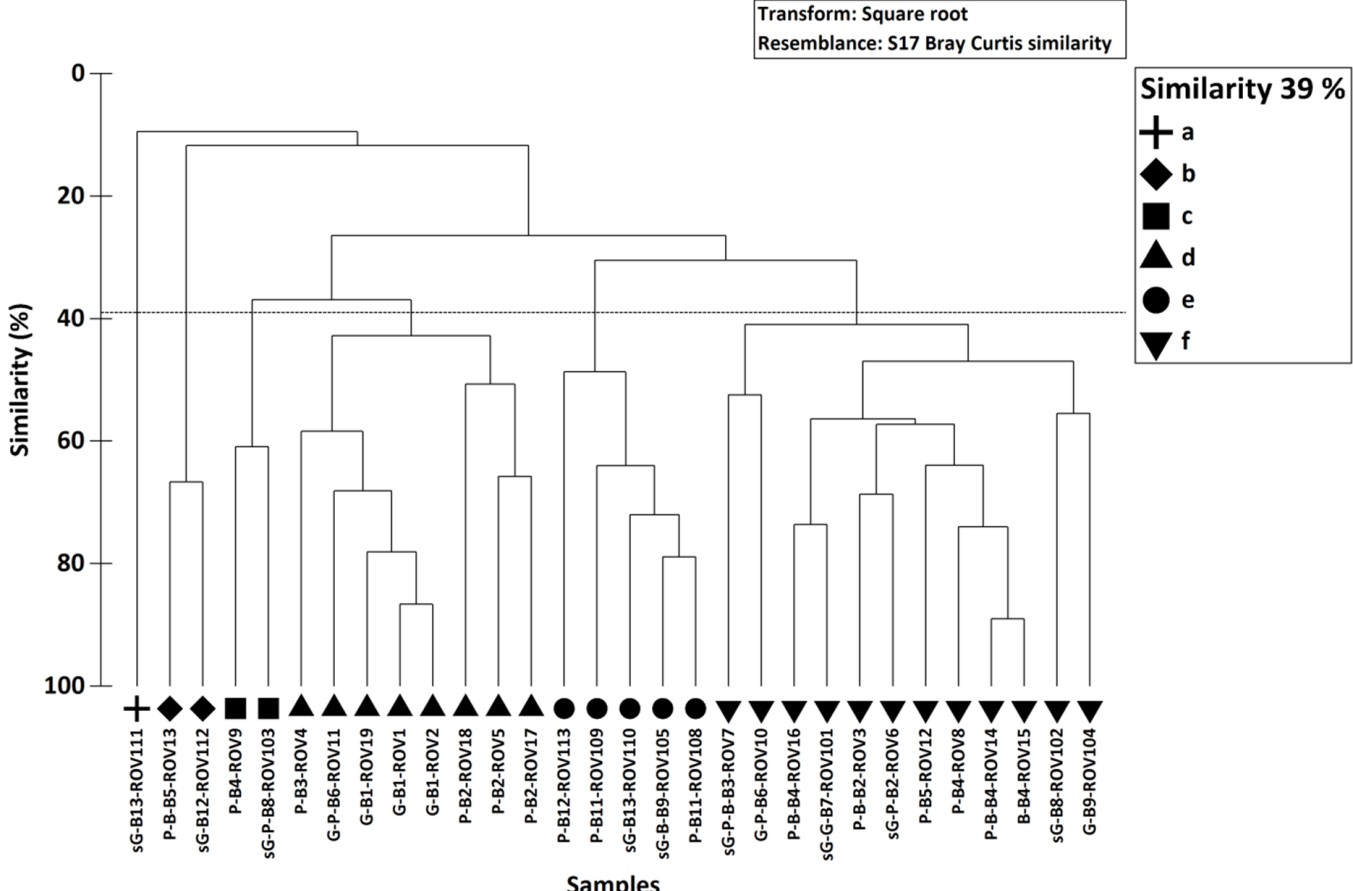

**Figure 8.** Cluster dendrogram showing the pattern of the 30 ROV sampling stations (motile and sessile taxa identified from the video and 35% of similarity) according to the Bray–Curtis similarity calculated using three classes of abundance ((1) rare; (2) common; (3) abundant and square root transformation).

**Table 6.** SIMPER analysis on sessile and motile taxa identified with the video from the mini ROV Seabotix with cumulative contribution (Cc in%) of the top species in the different groups identified by the cluster dendrogram analysis (Figure 8). (1): Group 1, . . . of Table 7. Group a (station ROV 111).

| Group b (1) | Cc (%) | Group c (2) | Cc (%) | Group d (3) | Cc (%) | Group e (4) | Cc (%) | Group f (5) | Cc (%) |
|---|---|---|---|---|---|---|---|---|---|
| *Alcyonium digitatum* | 100 | *Ophiothrix fragilis* | 33.66 | *Aequipecten opercularis* | 35.17 | *Abietinaria abietina* | 43.74 | *Flustrea* | 34.12 |
| - | - | *Alcyonium digitatum* | 61.14 | Hydrozoa | 55.69 | Sponges | 79.33 | Sponges | 50.47 |
| - | - | Hydrozoa | 80.57 | *Pagurus* spp. | 74.43 | *Alcyonium digitatum* | 89.79 | Hydrozoa | 63.66 |
| - | - | *Urticina felina* | 100 | *Asterias rubens* | 84.48 | *Nemertesia* | 97.67 | *Nemertesia* | 74.97 |
| - | - | - | - | *Urticina felina* | 90.15 | - | - | *Balanus crenatus* | 82.85 |
| - | - | - | - | - | - | - | - | *Spirobranchus* spp. | 89.61 |
| - | - | - | - | - | - | - | - | *Alcyonium digitatum* | 94.90 |

## 5. Discussion

The seabed of the EC is fashioned by high tidal hydrodynamics and is characterised by coarse sediments, especially in the subtidal areas. The management of this type of substrate is becoming more and more challenging for the marine blue economy in the EC due to the development of industries at sea such as marine aggregate extractions in the eastern channel paleo-valleys or the development of offshore wind farms, especially within French coastal waters where future wind farms sites have been allocated in the north of the Bay of Seine. The VIDEOCHARM study was a synoptic work in the central and the western part of the EC providing new quantitative data for the benthic communities at the end of spring, as both surveys were organised in June 2010 and 2011. Bad weather conditions affected the survey design, as most of the boxes sampled during the 2011 year (7 to 11) were relocated close to the Brittany coastline but remained in coarse circalittoral sediment, which is dominant in the EC [31].

### 5.1. EUNIS Classification

One of the aims of this study was to assess the diversity of coarse sediment offshore benthic habitats according to the EUNIS habitat classification [11,23,24,41]; https://www.eea.europa.eu/data-and-maps/data/eunis-habitat-classification-1 (accessed on 1 June 2022). Following the EUNIS 2012 classification, the sampling areas concerned two main habitats at Level 2—A5: sublittoral sediments and A4: circalittoral rocks and other substrata. Thus, the authors of [11] had modelled EUNIS map of the EC at this level of classification, and our study area corresponded in the eastern part (boxes 1 and 2) to the A5.14 habitat (circalittoral coarse sediment, i.e., MC32 Atlantic circalittoral coarse sediment of the EUNIS 2022 classification), while all the other boxes were classified as A5.15 habitat (deep circalittoral coarse sediment, i.e., MD2 offshore circalittoral coarse sediment of the EUNIS 2022 classification) (https://www.eea.europa.eu/data-and-maps/data/eunis-habitat-classification-1).

The EUNIS habitat classification was first published in the early 2000s and was developed according to the data available at the time. The classification was then better suited to describe shallow bays and estuarine environments (mobile muddy fine sediments) such as the Bay of Seine in the English Channel (42) rather than in coarser sediment habitats occurring in deeper areas such as those found in the central and western part of EC. The EUNIS habitat classification is now widely used to describe the diversity of marine benthic habitats of the European marine seabed [28,41].

The combination of approaches used in the VIDEOCHARM surveys (side scan sonar, video, photography, and sediment particle size and taxa identification) identified the offshore circalittoral coarse habitat to levels 5 and 6 of the EUNIS 2022 classification. The successive HCA proposed in this paper showed that most of the stations were always classified into the same cluster groups, whatever data source used, and only a few stations changed cluster groups according to the recorded taxa (Table 7). The cross analysis of these faunal groups with sediment data led to the allocation of the grab sampling stations to five EUNIS habitats.

**Table 7.** Summary information on the sediment types according to the photos (expert assessment) and sediment analyses for the 40 stations according to their location in the sampled boxes in June 2010 and June 2011. Positions of the stations according to the groups identified in the cluster dendrograms (see Figures 5–8 and Tables 3–6). The colours indicate the classification of each station into the five EUNIS 2022 habitat classifications. The numbers in red indicate that a station is classified in different groups according to the successive analyses; except MC4215 habitat in blue.

| Box | Station | Photos | Sediment Type | G 2 mm | G 2 + 1 mm | G 2 + 1 mm + Epifauna | ROV | EUNIS |
|---|---|---|---|---|---|---|---|---|
| 1 | 2 | Sandy Gravel | Sandy Gravel | 3 | 3 | 3 | 3 | MD3211 (A5.151) |
| | 3 | Sandy Gravel | Sandy Gravel | 3 | 3 | 3 | 3 | MD3211 (A5.151) |
| | 4 | Sandy Gravel | Sandy Gravel | 3 | 3 | 3 | 3 | MD3211 (A5.151) |
| | 5 | Sandy Gravel | Sandy Gravel | 3 | 3 | 3 | 3 | MD3211 (A5.151) |
| 2 | 6 | Sandy Gravel | Sandy Gravel | 3 | 3 | 3 | 5 | MD3211 (A5.151) |
| | 7 | Sandy Gravel | Sandy Gravel | 3 | 3 | 3 | 3 | MD3211 (A5.151) |
| | 8 | Sandy Gravel | Sandy Gravel | 3 | 3 | 3 | 3 | MD3211 (A5.151) |
| | 9 | Sandy Gravel | Sandy Gravel | 3 | 3 | 3 | 3 | MD3211 (A5.151) |
| 3 | 10 | Coarse sand | Sandy Gravel | 2 | 2 | 2 | 5 | MC3215 (A5.145) |
| | 11 | Sandy Gravel and Pebbles | Gravel | 3 | 3 | 3 | 5 | MD3211 (A5.151) |
| | 12 | Sandy Gravel | Gravel | 3 | 3 | 3 | - | MD3211 (A5.151) |
| | 14 | Sandy Gravel | Gravel | 3 | 3 | 3 | 5 | MD3211 (A5.151) |
| | 15 | Sandy Gravel | Gravel | 3 | 3 | 5 | - | MD3211 (A5.151) |
| | 16 | Sandy Gravel | Sandy Gravel | 3 | 3 | 3 | 5 | MD3211 (A5.151) |

**Table 7.** *Cont.*

| Box | Station | Photos | Sediment Type | G 2 mm | G 2 + 1 mm | G 2 + 1 mm + Epifauna | ROV | EUNIS |
|---|---|---|---|---|---|---|---|---|
| 4 | 18 | Sandy Gravel and Pebbles | Gravel | 3 | 3 | 3 | 5 | MD3211 (A5.151) |
| | 19 | Sandy Gravel and Pebbles | Gravel | 3 | 3 | 3 | 5 | MD3211 (A5.151) |
| | 20 | Gravel | Gravel | 5 | 3 | 5 | 5 | MC4215 (A5. 445) |
| | 21 | Sandy Gravel and Pebbles | Gravel | 5 | 3 | 3 | 2 | MC4215 (A5. 445) |
| 5 | 31 | Gravel | Gravel | 1 | 1 | 1 | - | MC4 (A5. 44) |
| | 32 | Sandy Gravel | Sandy Gravel | 1 | 1 | 1 | 1 | MC4 (A5. 44) |
| | 36 | Sandy Gravel and Pebbles | Gravel | 3 | 3 | 3 | - | MD3211 (A5.151) |
| | 38 | Sandy Gravel and Pebbles | Gravel | 3 | 3 | 5 | - | MD3211 (A5.151) |
| 6 | 22 | Gravelly Sand | Sandy Gravel | 3 | 3 | 3 | - | MD3211 (A5.151) |
| | 23 | Gravelly Sand | Sandy Gravel | 6 | 6 | 5 | 5 | MD3211 (A5.151) |
| | 24 | Gravelly Sand | Sandy Gravel | 3 | 3 | 4 | 5 | MD3211 (A5.151) |
| | 28 | Gravelly Sand | Sandy Gravel | 4 | 3 | 3 | 3 | MD3211 (A5.151) |
| 7 | 102 | Sandy Gravel and Pebbles | Sandy Gravel and Pebbles | 5 | 3 | 5 | - | MC4215 (A5. 445) |
| | 105 | Sandy Gravel and Pebbles | Sandy Gravel and Pebbles | 5 | 3 | 5 | 2 | MC4215 (A5. 445) |
| 8 | 108 | Sandy Gravel and Pebbles | Sandy Gravel and Pebbles | 4 | 4 | 2 | - | MC3212 (A5.142) |
| | 109 | Gravelly Sand and Pebbles | Sandy Gravel and Pebbles | 4 | 4 | 5 | - | MC3212 (A5.142) |
| 9 | 114 | Gravelly Sand and Pebbles | Sandy Gravel and Pebbles | 4 | 4 | 4 | 5 | MC3212 (A5.142) |

**Table 7.** *Cont.*

| Box | Station | Photos | Sediment Type | G 2 mm | G 2 + 1 mm | G 2 + 1 mm + Epifauna | ROV | EUNIS |
|---|---|---|---|---|---|---|---|---|
| 10 | 119 | Sandy Gravel and Pebbles | Sandy Gravel and Pebbles | 4 | 4 | 4 | 4 | MC3212 (A5.142) |
| | 120 | Gravelly Sand and Pebbles | Sandy Gravel and Pebbles | 4 | 4 | 4 | - | MC3212 (A5.142) |
| 11 | 125 | Gravelly Sand and Pebbles | Sandy Gravel and Pebbles | 4 | 4 | 4 | - | MC3212 (A5.142) |
| | 126 | Coarse Sand | Gravelly Sand | 2 | 2 | 2 | 4 | MC3215 (A5.145) |
| | 129 | Coarse Sand and Pebbles | Sandy Gravel and Pebbles | 4 | 4 | 4 | 4 | MC3212 (A5.142) |
| | 130 | Gravelly Sand | Sandy Gravel and Pebbles | 4 | 4 | 4 | 4 | MC3212 (A5.142) |
| 13 | 131 | Coarse Sand | Gravelly Sand | 2 | 4 | 4 | 6 | MC3215 (A5.145) |
| | 133 | Gravelly Sand | Sandy Gravel and Pebbles | 4 | 4 | 4 | 1 | MC3212 (A5.142) |
| | 134 | Gravelly Sand | Sandy Gravel and Pebbles | 4 | 4 | 4 | 4 | MC3212 (A5.142) |

MC3215 (A5.145). *Branchiostoma lanceolatum* in Atlantic circalittoral coarse sand with shell gravel. Only three stations (10 in Box 3, 126 in Box 126 and 131 in Box 13) on coarse sand corresponded to this habitat.

MD3211 (A5.151). *Glycera lapidum*, *Thyasira* spp. and *Amythasides macroglossus* in offshore circalittoral gravelly sand. A total of 21 stations corresponded to this habitat. They were located the eastern boxes 1 to 4, as well as in boxes 5 and 6 located in the north of the Channel Islands.

MC4 (A5.44) Circalittoral mixed sediment. Two stations, 31 and 32, from Box 5 corresponded to this habitat with a very low number of taxa permitting to identify the habitat only at level 2 of the classification.

MC3212 (A5.142) *Mediomastus fragilis*, *Lumbrineris* spp. and venerid bivalves in Atlantic circalittoral coarse sand or gravel. This habitat corresponded to 10 stations all located in boxes 8 and 13 in the western part of the EC.

MC4215 (A5.445) *Ophiothrix fragilis* and/or *Ophiocomina nigra* brittle star beds on circalittoral-mixed sediment. Four stations, two in Box 4 (20 and 20) and two in Box 7 (102 and 105), were assigned to this habitat characterised by the presence of the brittle star *Ophiothrix fragilis*.

Both MC3212 (A5.142) and MC4215 (A5.445) were dominant in the central part of the Bay of Seine [42]. Moreover, gravel and pebble sediment were largely distributed in

the eastern part of the EC [25,26]. Nevertheless, these last authors did not distinguish the EUNIS habitat in their study.

On the Dieppe-Le Tréport (DLT) Offshore Wind Farm (French coastal part of the eastern EC), a sampling strategy was developed in 2014–2016 to establish a 'Before' state for the sediment and macrofauna. The coarse sediment assemblage sampled on sandy gravel and gravelly sand corresponds to two EUNIS habitats MC3212 characterised by molluscs with large biomass such as *Glycymeris* and *Polititapes rhomboids* [39]. Some stations are dominated by the cephalocordate *Branchiostoma lanceolatum*, yielding an assemblage corresponding rather to the EUNIS habitat MC3215, which was largely represented in coastal coarse sediment along the Brittany coast such as in the Bay of Morlaix [9,43].

Our study is the first to describe in such detail the patterns of distribution of the offshore coarse sediment habitats. Two habitats dominated the surveyed area, the Sandy Gravel MD3211 in the eastern part of the EC and the Gravelly Sand and Pebbles MC3212 in the western EC. The MC4215 (dense bed of *O. fragilis*) was rare in the offshore coarse sediment and was more largely distributed in the Bay of Seine [42]. Our study completed the review provided by [44].

*5.2. New Quantitative Data on the Coarse Sediment of the Central Part of the English Channel*

The taxonomic richness and the mean abundance per m$^2$ of the five EUNIS habitats reported in the central part of the EC were compared with values found on the sandy gravel and gravelly sand found elsewhere in the English Channel (Table 8) [39,43,45–54]. The taxonomic richness appeared to be correlated to the sampling area with the highest values corresponding to the largest sampling areas. Nevertheless, the low TR reported for the MC3215 (coarse sand) and MC4 (gravel) were among the smallest values recorded in similar coarse sediment habitats in the EC. The TR for the sandy gravel MD3211 and the sandy gravel MC3212 habitats, respectively 233 and 164, were however within the range of values found in the samples collected along the French coast of the eastern part of the EC and for similar sampling surfaces (from 147 in June from the future Offshore Wind Farm of Courseulles-sur-Mer to 277 in September/October from the future Offshore Wind Farm of Dieppe-Le-Tréport). The TR of the MC4215 habitat corresponding to coarse sand sediment with the presence of *Ophiothrix fragilis* was low; similarly, the abundance value (316 individuals per m$^2$) remained moderate in comparison with a similar habitat of the Bay of Seine where large populations of *O. fragilis* have been observed (>5000 individuals per m$^2$) [13,55].

As reported in Table 8, abundances on coarse sediments in the EC varied from a minimum of 192 individuals per m$^2$ to a maximum of 4590 individuals per m$^2$. Values were lowest in the western EC (coarse sand of the Bay of Morlaix; 192 individuals per m$^2$) compared to other sites in the eastern EC. Our study reported abundances between 223 and 800 individuals per m$^2$ in the central part of the EC, which were among the lowest reported in the EC for similar sediment types. The depths of the five habitats sampled during both our surveys were higher than those of other coastal studies (Table 8); moreover, the primary production and organic matter fluxes between the water column and the sea bottom were lower than those observed near the coast where nutrient fluxes coming from rivers increased the primary production [29,55].

In summary, in the offshore coarse sediment of the central part of the EC, the taxonomic richness of both extensive coarse habitats was on the same order of magnitude as the similar coarse sand habitat of the English Channel, whilst the abundances were lower than those found near the shallow coastal coarse sediment, especially in the Bay of Seine and in the eastern EC. These low abundance values, and probably low biomasses, of the macrofauna in the deep central zone of the EC can be related to a low nutrient input and primary production and low pelagic fluxes from the water column to the benthic habitat.

**Table 8.** Univariate indices and grab sampling details for the accountable macrofauna from the coarse sediment (sieving on 1 mm): coarse sand (Cs) sandy gravel (sG), gravelly sand (gS), gravel (G), pebbles (P) and O (*Ophiothrix fragilis*) for the English Channel. (wEC: west English Channel; eEC: east English Channel); UK: United Kingdom; Ss: sampling surface in m$^2$; D: depth in m; S: sediment type; TR: taxonomic richness: A: abundance, individual number per m$^2$ (in part from 39).

| | | Site | Month | Year | Ss | D | S | TR | A (m$^2$) | Reference |
|---|---|---|---|---|---|---|---|---|---|---|
| wEC | UK | West EC | - | - | - | - | sG and gS | - | 390 | MESL, 1999 [46] |
| | France | Morlaix | Each month | 1977–1980 | 32.5 | 17 | Cs | 181 | 192 | Dauvin, 1988 [43] |
| eEC | France | Dieppe | - | 1996–1997 | 0.9 | 15 | sG | 50 | 1.940 | Desprez, 2000 [47] |
| | | | - | 1996–2001 | 0.8 | 15 | sG | 50 | 2.394 | Desprez et al., 2010 [48] |
| | | Dieppe-Le Tréport | September/October | 2014–2016 | 14 | 12–25 | sG | 277 | 2.989 | Pezy and Dauvin, 2021 [39] |
| | | | February/March | | 24 | | gS | 224 | 1.605 | |
| | | Bay of Seine | June–August | 2007 | 19 | 38–50 | sG | 198 | 1.309 | Lozach and Dauvin, 2012 [13] |
| | | PER Granulats du Havre | February | 2012 | 2.5 | 16–22 | sG | 117 | 777 | Pezy et al., 2021 [49] |
| | | | February | 2021 | 6 | | sG and gS | 157 | 1.219 | Pezy et al., 2021 [49] |
| | | Courseulles-sur-Mer | June | 2009 | 8.1 | 22–28 | sG | 147 | 377 | In Vivo, 2013 [50] |
| | | | March | 2020 | 4.5 | 22–30 | sG | 182 | 3.303 | Raoux et al., 2021 [51] |
| | | | March | 2021 | 5.4 | 22–30 | sG | 159 | 2.008 | |
| | UK | St Catherine | - | - | - | - | sG and gS | - | 4.590 | MESL, 1996 [45] |
| | | West Bassurelle | - | - | - | - | sG and gS | - | 932 | MESL, 1999 [46] |
| | | Folkestone | - | - | - | - | Cs | - | 3.051 | Newell et al., 2001 [52] |
| | | Isle of Wight | March and September | 1999 | 26.2 | >10 | Cs | 316 | 998 | Newell et al., 2004 [53] |
| | | Hastings | - | - | - | - | sG | - | 2.000 | Cooper et al., 2007 [54] |
| | Offshore central and western part of the EC | | June | 2010–2011 | 1.5 | 64–95 | Cs | 33 | 223 | This study |
| | | | | | 1 | 88–96 | G | 7 | 434 | |
| | | | | | 10.5 | 45–80 | sG | 233 | 800 | |
| | | | | | 5 | 73–101 | sG and P | 164 | 380 | |
| | | | | | 2 | 54–67 | sG and P + O | 80 | 316 | |

### 5.3. Information Gained from the Seven Benthic Habitats Sampling Methods

The originality of our study was the multidisciplinary approach used to assess the best method to identify the different circalittoral coarse sediment habitats. Both imagery (side scan sonar, photography and ROV) and direct approaches (grab sampling with benthic identifications for (epifauna and endofauna) on two sieving mesh, and particle size analyses) were developed and combined.

Even if sonar profiles did not fully cover all box areas, they gave more spatial information than grab samples and ROV video footage. Sedimentary and morphological structures are clearly highlighted (when present) to estimate rapidly possible habitats. Cross-analysis of the side scan sonar profiles with information obtained through the analysis of the grab samples should allow for semi-automatic spatial classification. However, this approach suffers in the present study from the lack of cross-information over the acoustic responses highlighted on survey boxes. The profiles showed a high variability of acoustic responses (in particular in boxes 3 and 4; Figures 4 and 9) that would require a high sampling effort to perform an exhaustive classification. At the scale of the EC, acoustic profiles also showed high frequency variations. These latter could not have been preliminarily taken into account, i.e., before the surveys, due to lack of associated sedimentary information.

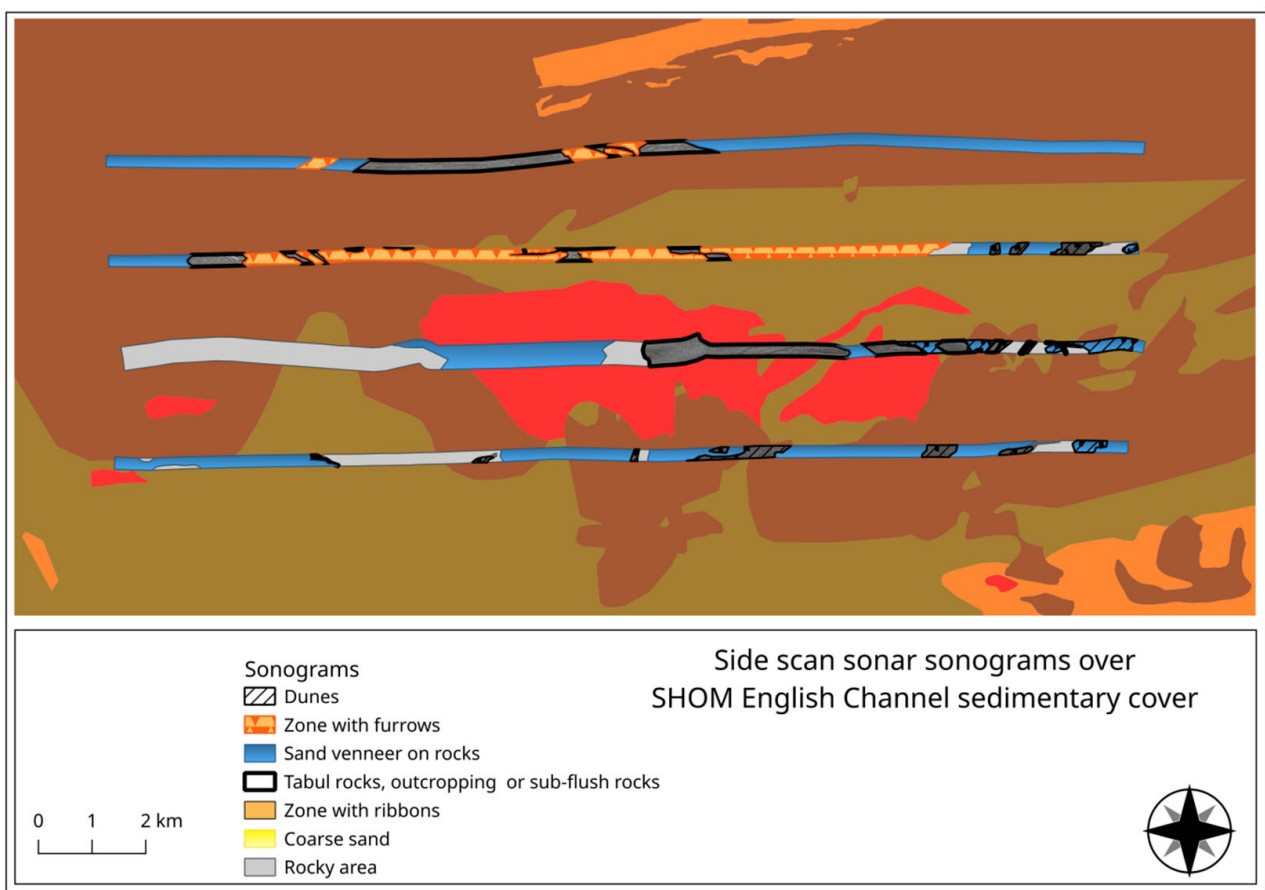

**Figure 9.** Side scan sonar profiles collected during the VIDEOCHARM surveys in June 2010 (box 4) over the SHOM sedimentary cover in the English Channel.

For example, the comparison of sonar responses with efficient EC sedimentary cover available at SHOM (data.shom.fr) showed a higher variability of sonograms (Figure 9). However, this approach can be applied to smaller areas in the frame of particular objectives, such as preliminary state establishment of environmental context before installation of marine renewable energy structures (wind farms or water turbine farms). Figure 10 shows the EUNIS habitat map from EUSeaMap product (https://www.emodnet-seabedhabitats.eu, accessed on 1 April 2022) with the VIDEOCHARM side scan sonar survey lines overimposed. VIDEOCHARM data sets confirm our EUNIS classification for boxes 1 and 2. Box 11 partially covers both high- and a moderate-energy circalittoral areas according to EUSeaMap. The difference between these two areas is highlighted on the side scan sonar profile with the presence of megadunes on the highest energy part (the two northwest profiles of Box 11). This is also the case for Box 11, entirely located on a high-energy circalittoral rock area, which shows megadunes both with sonar profile data and ROV. Moreover, it will be interesting in the future to integrate our VIDEOCHARM data in the EUSeaMap.

First, coding was developed for side-scan sonar images to describe seabed morphology (Table 1). The authors of [28,56] had suggested to add at level 2 of the EUNIS classification, for the circalittoral rocks and the sub-littoral sediment, additional information on the forms of the rocks and the presence of ribbon and dune bed forms (including dune wavelength), and to note evidence of anthropogenic activity. Such longitudinal furrows and sand ribbons and rocky reef had been previously observed in the central part of the EC [57,58]. This supplementary data can help to better understand the heterogeneity of such a deep habitat, as observed by [11,23], for coarse sediment offshore from the English side of the EC.

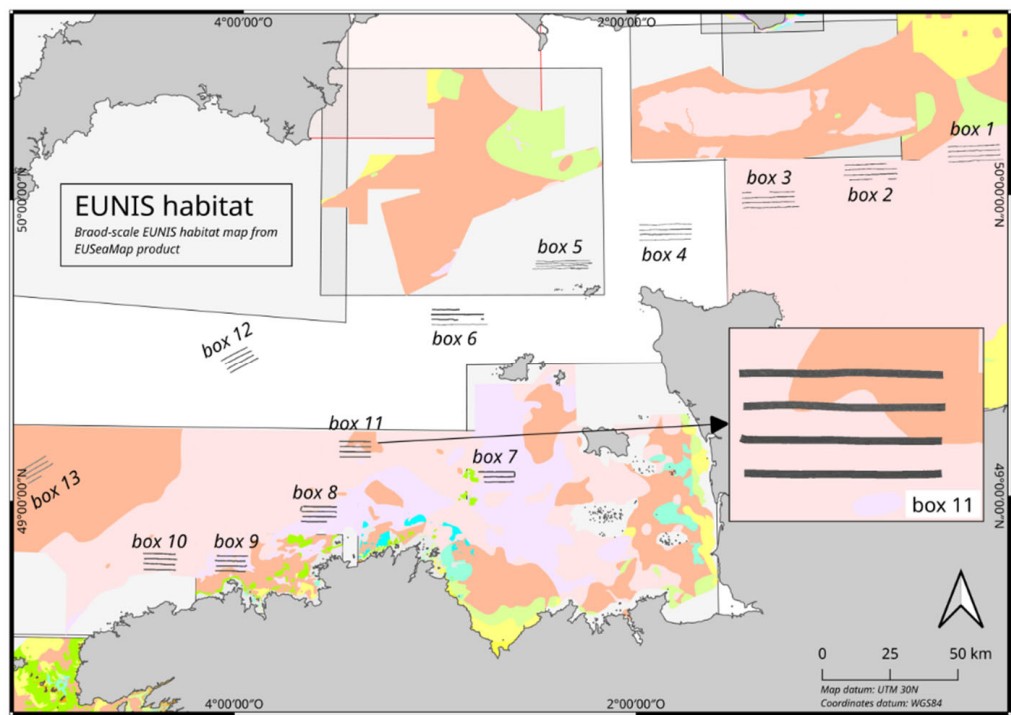

**Figure 10.** EUNIS habitat map from the library of marine habitats maps in European waters (EMODnet seabed habitats). VIDEOCHARM side scan sonar survey lines are shown with corresponding box number.

The sessile epifauna appeared relatively homogenous in this type of sandy gravel and gravel sediment at the scale of the sampling area and was not a driver in the classification of the offshore coarse sediment habitats of the central part of the EC. Nevertheless, the sampling surface remained small (0.5 m²) and was probably insufficient for good insight into the diversity of the sessile fauna and could not detect the western–eastern impoverishment of these fauna identified at the scale of the whole EC by [10]. These authors have shown a gradual disappearance of sessile species from the western part to the eastern part of the EC in relation to the annual amplitude of the sea water temperature, which was higher in the east part than in the western part. Moreover, about 3000 sampling stations with a Rally du Baty dredge were used to describe this general distribution pattern of the sessile fauna in the whole EC.

As highlighted later by Foveau [25,26] for the benthic habitats of the eastern part of the EC, uncountable taxa, belonging to the sessile epifauna, were logically mainly associated with pebbled areas, located offshore the 'Pays de Caux' near the Antifer Cape and the Dover Strait. The location of diversity hotspots (total or sessile epifauna only) coincided with the distribution of large gravel and pebble sediment particles.

Sessile fauna taxa recorded in the 40 Hamon grab sampling stations in gravel and pebble sediments during both VIDEOCHARM surveys included 125 taxa which represented 31% of the total diversity. Sessile epifauna are often neglected in research projects on the soft-bottom benthic community despite representing an important source of diversity. Nevertheless, as numerous taxa are colonials, only their presence can be accounted for and used in statistical analyses.

ROV video footage was useful in accounting for the large vagile and sessile epifauna, which were spatially dispersed and not sampled by the Hamon grab, which only collects a relatively small surface of 0.25 m². This method gave valuable supplementary observations, providing information for the interpretation of the acoustic profiles on the nature of surficial sediment and presence of dispersed or dense epifauna such as the *Ophiothrix fragilis*. Some taxa were identified only on video footage such as the bryozoans *Alcyonidium* spp.,

the Cnidaria *Alcyonium digitatum*, and six large echinoderms. The epifauna assemblage pattern observed from the video footage was comparable to those of the endofauna. The videos were also useful as a communication tool to illustrate the results to non-scientists such as fishermen, stakeholders and marine environment managers.

Expert description on board of the sediment collected by the grab gave good information on the sediment type (the photo permitted to store a visual memory of the grab to verify the expert judgement if necessary). Granulometric analyses gave supplementary information on fine particles (very low content in such coarse sediment) and the respective percentage of particles between pebbles, gravel and sands as sediment components.

In the early benthic habitat studies in the EC, scientists used dredge [28], and then later in the first quantitative studies, they used grabs such as the Hamon grab which allowed the collection of sufficient volume of coarse sediment [13,25,26,55]. These authors then used a 2 mm sieve mesh to retain the macrofauna. Our results hence gave, for the first time at the scale of 80 Hamon grabs sampled in the coarse sediment on the French side of the EC, the taxonomic richness and abundances of the macrofauna retained on a 2 mm and on a 1 mm mesh size because there are many samples in the UK waters on 1 mm in the EC. Considering the taxonomic diversity, most of the species (84%) of the taxa) had been retained by a 2 mm mesh sieve, and only the remaining 16% (mainly small polychaetes and amphipods) passed through the 2 mm mesh sieve and had been retained by a 1 mm mesh size. Thus, the increase in taxonomic richness was weak with the double sieving on 2 and 1 mm. Then, considering the abundances, more than one third of the total number of individuals passed through on a 2 mm mesh size and was retained by a 1 mm mesh size. These small individuals had an important consequence on the ecological status of the benthic habitat (see Table 2), which showed better ecological status expressed by J' and H' when they were assessed with the macrofauna sampled on a 1 mm mesh sieve. Nevertheless, as underlined by [13], in most of the investigations of the French side of the EC, a 2 mm mesh sieve had been frequently used because more than 95% of macrobenthic biomass was generally retained. In summary, improvements of using a 1 mm mesh sieve mesh are highly important for abundances but are of low or moderate importance for the biomasses and taxonomic richness.

Nevertheless, based on these considerations, it is recommended to sieve the macrofauna on a 1 mm mesh for the entire area of the EC where grab (Hamon or others) can be used.

### 5.4. Perspectives on the Future of the EC Marine Management

The French government has announced in early March 2022 a 'target 40 gigawatts of marine renewable energy in service in 2050', which is the equivalent of about fifty offshore wind farms (OWF) to be installed in the French Metropolitan coastal waters. Four OWFs were under construction along the French side of the EC at a distance between 15 and 20 km from the coastline [59,60]. In the future, there is a plan to build OWF at a greater distance from the coast (around 50 km), similar to the area prospected during the VIDEOCHARM surveys offshore the Bay of Seine where two new OWF sites have been investigated [59]. Similarly, aggregate extraction areas are planned at a greater distance from the coast, such as the area located to the north of the Bay of Seine [13]. The VIDEOCHARM data could therefore be used as a reference point for the macrofauna compartment as well as an important source of information to carry out the environmental impact assessment needed prior to any industrial development at sea. Data on the biomass of the benthic macrofauna should be completed by data on other biological components such as phytoplankton, zooplankton, suprabenthos and fishes to understand fluxes between these components (e.g., by studying fish stomach contents) to provide an ecosystem approach of such an offshore system in the central part of the EC. The development of the marine renewable energy structure in the EC will be for the researchers an opportunity to increase the knowledge of the functioning of such offshore coarse sediment, which has remained, once again, poorly investigated [59–61].

**Author Contributions:** Conceptualization J.-C.D., S.L., E.P. and A.T.; methodology, J.-C.D., S.L., E.P. and A.T.; software, E.P. and J.-P.P.; validation, all the co-authors.; formal analysis, J.-P.P. and E.P.; investigation, J.-C.D., S.L., E.P. and A.T.; resources, J.-C.D. and E.P.; data curation, J.-C.D. and J.-P.P.; writing—original draft preparation, J.-C.D.; writing—review and editing, all the co-authors; visualization, J.-C.D.; supervision, J.-C.D.; project administration, J.-C.D.; funding acquisition, J.-C.D. All authors have read and agreed to the published version of the manuscript.

**Funding:** This research received financial support from the INTERREG IV A France (Channel)–England cross-border European cooperation programme, co-financed by the European Regional Development Fund as part of the CHannel integrated Approach for marine Resource Management (CHARM) Phase 3 project.

**Institutional Review Board Statement:** Not applicable.

**Informed Consent Statement:** Not applicable.

**Data Availability Statement:** The raw data should be available.

**Acknowledgments:** This work was carried out under the INTERREG IV A France (Channel)–England cross-border European cooperation programme, co-financed by the European Regional Development Fund as part of the CHannel integrated Approach for marine Resource Management (CHARM) Phase 3 project. The authors thank the crews of RV 'Côtes de la Manche, as well as all the scientists and students who helped during the cruises, especially R. Abraham, D. Malengros, A. Baffreau, and A. Foveau who help the identifications of species and taxa. The authors also thank the two reviewers for their attentive lecture and their useful suggestions to amend the first version of this paper.

**Conflicts of Interest:** The authors declare no conflict of interest.

## Appendix A

Appendix A. Details of the sampling effort during the VIDEOCHARM surveys in June 2010 (boxes 1 and 6) and June 2011 (boxes 7 to 13) in the central part of the English Channel. ST: station; RE: replicate; ROV: remote observatory vehicle; B: benthic samples; H1: Hamon grab replicate 1, H2: Hamon grab replicate 2, G: Hamon grab for granulometry. Sediment photos: sediment assessment from expert judgement during the survey.

| DATE | Box | Code | ST | RE | LATITUDE | LONGITUDE | Depth | SEDIMENT PHOTOS |
|---|---|---|---|---|---|---|---|---|
| 05/06/2010 | 1 | ROV1 | 2 | 1 | 50.14906667 | −0.20678333 | 49.64 | |
| 05/06/2010 | 1 | B2H1 | 2 | H1 | 50.14915 | −0.20561667 | 46.6 | gravel, coarse sand: sandy gravel |
| 05/06/2010 | 1 | B2H2 | 2 | H2 | 50.14915 | −0.20561667 | 46.6 | coarse sand, gravel: sandy gravel |
| 05/06/2010 | 1 | B2G | 2 | G | 50.14915 | −0.20561667 | 46.6 | coarse sand, gravel and pebbles: sandy gravel |
| 05/06/2010 | 1 | ROV1bis | 2 | 2 | 50.14933333 | −0.20676667 | 48.93 | |
| 05/06/2010 | 1 | B3H1 | 3 | H1 | 50.13373333 | −0.14618333 | 50.12 | coarse sand, gravel and shells: sandy gravel |
| 05/06/2010 | 1 | B3H2 | 3 | H2 | 50.13388333 | −0.14575 | 50.07 | coarse sand, gravel, pebbles and shells: sandy gravel |
| 05/06/2010 | 1 | B3G | 3 | G | 50.134 | −0.14526667 | 50.03 | gravel, coarse sand: sandy gravel |
| 05/06/2010 | 1 | B4G | 4 | G | 50.16738333 | −0.09878333 | 46.61 | coarse sand, gravel and pebbles: sandy gravel |
| 05/06/2010 | 1 | B4H1 | 4 | H1 | 50.16755 | −0.09776667 | 47.61 | coarse sand, gravel and pebbles: sandy gravel |
| 05/06/2010 | 1 | B4H2 | 4 | H2 | 50.16781667 | −0.0963 | 47.54 | coarse sand, gravel and pebbles: sandy gravel |
| 05/06/2010 | 1 | ROV2 | 5 | 1 | 50.13233333 | −0.28108333 | 46.55 | |
| 05/06/2010 | 1 | B5G | 5 | G | 50.13206667 | −0.28183333 | 43.88 | gravel, coarse sand, shells: sandy gravel |
| 05/06/2010 | 1 | B5H1 | 5 | H1 | 50.13206667 | −0.28183333 | 44.88 | coarse sand, gravel and pebbles: sandy gravel |
| 05/06/2010 | 1 | B5H2 | 5 | H2 | 50.13206667 | −0.28183333 | 43.88 | coarse sand, gravel: sandy gravel |
| 06/06/2010 | 2 | ROV3 | 6 | 1 | 50.11665 | −0.70995 | 54.14 | |
| 06/06/2010 | 2 | B6H1 | 6 | H1 | 50.11688333 | −0.71191667 | 53.14 | pebbles, gravel and coarse sand: sandy gravel |
| 06/06/2010 | 2 | B6G | 6 | G | 50.11651667 | −0.71616667 | 53.14 | pebbles, gravel and coarse sand: <span style="color:red">sandy gravel</span> |
| 06/06/2010 | 2 | B6H2 | 6 | H2 | 50.11663333 | −0.71723333 | 52.14 | gravel, coarse sand, shells: sandy gravel |
| 06/06/2010 | 2 | B7H1 | 7 | H1 | 50.08251667 | −0.6385 | 48.14 | coarse sand, gravel, pebbles and shells: sandy gravel |
| 06/06/2010 | 2 | B7G | 7 | G | 50.08265 | −0.64048333 | 48.14 | gravel, coarse sand: sandy gravel |
| 06/06/2010 | 2 | B7H2 | 7 | H2 | 50.08313333 | −0.64466667 | 48.14 | coarse sand, gravel: sandy gravel |
| 06/06/2010 | 2 | B8G | 8 | G | 50.06643333 | −0.64038333 | 50.14 | gravel, coarse sand: sandy gravel |
| 06/06/2010 | 2 | B8H1 | 8 | H1 | 50.06623333 | −0.64315 | 50.14 | coarse sand and gravel: sandy gravel |
| 06/06/2010 | 2 | B8H2 | 8 | H2 | 50.0664 | −0.64425 | 50.14 | gravel, coarse sand: sandy gravel |
| 06/06/2010 | 2 | B9G | 9 | G | 50.06808333 | −0.7975 | 57.14 | gravel, coarse sand: sandy gravel |
| 06/06/2010 | 2 | B9H1 | 9 | H1 | 50.06765 | −0.79716667 | 58.14 | pebbles and gravel: gravelly pebble |
| 06/06/2010 | 2 | B9H2 | 9 | H2 | 50.06765 | −0.79725 | 58.14 | pebbles and gravel: gravelly pebble |
| 06/06/2010 | 2 | ROV4 | 9 | 1 | 50.0684 | −0.79653333 | 58.14 | |
| 06/06/2010 | 2 | ROV5 | 9 | 2 | 50.0684 | −0.79653333 | 58.14 | |
| 07/06/2010 | 3 | ROV6 | 10 | 1 | 49.98395 | −1.16641667 | 62.46 | |
| 07/06/2010 | 3 | B10H1 | 10 | H1 | 49.98326667 | −1.16828333 | 63.93 | coarse sand (mineral) |
| 07/06/2010 | 3 | B10H2 | 10 | H2 | 49.98326667 | −1.16828333 | 63.99 | coarse sand (mineral) |
| 07/06/2010 | 3 | B10G | 10 | G | 49.98326667 | −1.16828333 | 64.01 | coarse sand (mineral) |
| 07/06/2010 | 3 | B11H1 | 11 | H1 | 50.00111667 | −1.18013333 | 55.43 | pebbles, sand and gravel: sandy gravel and pebble |
| 07/06/2010 | 3 | B11H2 | 11 | H2 | 50.00178333 | −1.18365 | 57.5 | pebbles, sand and gravel: sandy gravel and pebble |
| 07/06/2010 | 3 | B11G | 11 | G | 49.99986667 | −1.17986667 | 55.59 | pebbles, sand and gravel: sandy gravel and pebble |
| 07/06/2010 | 3 | B12H1 | 12 | H1 | 49.99875 | −1.27925 | 66.9 | pebbles with incrusting fauna |
| 07/06/2010 | 3 | B12G | 12 | G | 49.99833333 | −1.28043333 | 67.93 | gravel and coarse sand: sandy gravel |
| 07/06/2010 | 3 | B12H2 | 12 | H2 | 49.99783333 | −1.28195 | 68.88 | gravel and coarse sand: sandy gravel |
| 07/06/2010 | 3 | B13G | 13 | G | 49.99888333 | −1.34813333 | 95.26 | gravel, coarse sand and mud: sandy gravel |
| 07/06/2010 | 3 | B14G | 14 | G | 50.00106667 | −1.37768333 | 74.23 | pebbles, sand and gravel: sandy gravel |
| 07/06/2010 | 3 | B14H1 | 14 | H1 | 50.00303333 | −1.3803333 | 67.16 | pebbles, sand and gravel: sandy gravel |
| 07/06/2010 | 3 | B14H2 | 14 | H2 | 50.0036 | −1.38125 | 68.13 | pebbles, sand and gravel: sandy gravel |
| 07/06/2010 | 3 | ROV7 | 14 | 1 | 50.00116 | −1.37737 | 74.25 | |
| 07/06/2010 | 3 | B15H1 | 15 | H1 | 49.999 | −1.38511667 | 75.76 | gravel and coarse sand: sandy gravel |
| 07/06/2010 | 3 | B15G | 15 | G | 49.99941667 | −1.3893 | 70.66 | pebbles, sand and gravel: sandy pebble |
| 07/06/2010 | 3 | B15H2 | 15 | H2 | 49.9992 | −1.38666667 | 74.62 | coarse sand and gravel: sandy gravel |
| 07/06/2010 | 3 | B16H1 | 16 | H1 | 49.99925 | −1.39091667 | 70.28 | pebbles, sand and gravel: sandy gravel |
| 07/06/2010 | 3 | B16H2 | 16 | H2 | 49.99963333 | −1.39353333 | 67.14 | pebbles, sand and gravel: sandy gravel |
| 07/06/2010 | 3 | B16G | 16 | G | 49.99925 | −1.39091667 | 70.28 | pebbles, sand and gravel: sandy gravel |

| DATE | Box | Code | ST | RE | LATITUDE | LONGITUDE | Depth | SEDIMENT PHOTOS |
|---|---|---|---|---|---|---|---|---|
| 08/06/2010 | 4 | ROV8 | 17 | 1 | 49.91855 | −1.814 | 72.28 | |
| 08/06/2010 | 4 | B17G | 17 | G | 49.8834 | −1.86686667 | 68.49 | pebbles |
| 08/06/2010 | 4 | B18G | 18 | G | 49.8835 | −1.92143333 | 68.75 | pebbles, gravel and coarse sand: sandy gravel and pebbles |
| 08/06/2010 | 4 | B18H1 | 18 | H1 | 49.8835 | −1.92143333 | 68.75 | pebbles, gravel and coarse sand: sandy gravel and pebbles |
| 08/06/2010 | 4 | B18H2 | 18 | H2 | 49.8835 | −1.92143333 | 68.75 | pebbles, gravel and coarse sand: sandy gravel and pebbles |
| 08/06/2010 | 4 | B19H1 | 19 | H1 | 49.93256667 | −1.92101667 | 77.45 | pebbles, gravel: sandy gravel and pebbles |
| 08/06/2010 | 4 | B20H1 | 20 | H1 | 49.93568333 | −1.7767 | 73.2 | gravel, sand: sandy gravel + *Ophiothrix fragilis* |
| 08/06/2010 | 4 | B20H1 | 20 | H2 | 49.93568333 | −1.7767 | 73.2 | clean gravel: gravel |
| 08/06/2010 | 4 | B20G | 20 | G | 49.93568333 | −1.7767 | 73.2 | clean gravel: gravel |
| 08/06/2010 | 4 | B21H1 | 21 | H1 | 49.91535 | −1.69925 | 67.66 | clean pebbles, gravel and sand: sandy gravel and pebbles |
| 08/06/2010 | 4 | B21G | 21 | G | 49.91535 | −1.70026667 | 67.68 | clean pebbles, gravel and sand: sandy gravel and pebbles |
| 08/06/2010 | 4 | B21H2 | 21 | H2 | 49.926 | −1.7023 | 69.78 | clean pebbles, gravel and sand: sandy gravel and pebbles |
| 08/06/2010 | 4 | ROV9 | 21 | 1 | 49.926 | −1.7023 | 69.78 | |
| 11/06/2010 | 6 | B22G | 22 | G | 49.6074 | −2.74438333 | 66.62 | sand and gravel: gravelly sand |
| 11/06/2010 | 6 | B22H1 | 22 | H1 | 49.60488333 | −2.74753333 | 66.56 | sand and gravel: gravelly sand |
| 11/06/2010 | 6 | B22H2 | 22 | H2 | 49.60386667 | −2.74903333 | 66.48 | sand and gravel: gravelly sand |
| 11/06/2010 | 6 | B23H1 | 23 | H1 | 49.60408333 | −2.81343333 | 68.94 | sand and gravel: gravelly sand |
| 11/06/2010 | 6 | B23H2 | 23 | H2 | 49.60203333 | −2.81508333 | 67.88 | sand and gravel: gravelly sand |
| 11/06/2010 | 6 | B23G | 23 | G | 49.60851667 | −2.81373333 | 68.68 | sand and gravel: gravelly sand |
| 11/06/2010 | 6 | ROV10 | 23 | 1 | 49.61246667 | −2.81246667 | 69.08 | |
| 11/06/2010 | 6 | ROV10bis | 23 | 2 | 49.60955 | −2.8122 | 68.54 | |
| 11/06/2010 | 6 | B24G | 24 | G | 49.60771667 | −2.88858333 | 70.28 | sand and gravel: gravelly sand |
| 11/06/2010 | 6 | B24H1 | 24 | H1 | 49.60655 | −2.87145 | 70.21 | sand and gravel: gravelly sand |
| 11/06/2010 | 6 | B24H2 | 24 | H2 | 49.6097 | −2.89128333 | 70.2 | sand and gravel: gravelly sand |
| 11/06/2010 | 6 | B25G | 25 | G | 49.6065 | −2.89561667 | 71.19 | sand and gravel: gravelly sand |
| 11/06/2010 | 6 | B26G | 26 | G | 49.60831667 | −3.00031667 | 73.29 | sand and gravel: gravelly sand |
| 11/06/2010 | 6 | B27G | 27 | G | 49.62575 | −2.96721667 | 74.4 | sand and gravel: gravelly sand |
| 11/06/2010 | 6 | B28H1 | 28 | H1 | 49.62561667 | −2.87531667 | 71.64 | sand and gravel: gravelly sand |
| 11/06/2010 | 6 | B28H2 | 28 | H2 | 49.62648333 | −2.874 | 71.69 | sand and gravel: gravelly sand |
| 11/06/2010 | 6 | B28G | 28 | G | 49.62725 | −2.87323333 | 71.69 | sand and gravel: gravelly sand |
| 11/06/2010 | 6 | ROV11 | 28 | 1 | 49.62463333 | −2.874 | 72.32 | |
| 12/06/2010 | 5 | ROV12 | 29 | 1 | 49.79016667 | −2.3665 | 75.14 | |
| 12/06/2010 | 5 | ROV12bis | 29 | 2 | 49.79016667 | −2.3665 | 79.14 | |
| 12/06/2010 | 5 | B29G | 29 | G | 49.79433333 | −2.38095 | 78.14 | pebbles + incrusting fauna |
| 12/06/2010 | 5 | B30G | 30 | G | 49.79945 | −2.43516667 | 89.14 | pebbles + incrusting fauna |
| 12/06/2010 | 5 | B31G | 31 | G | 49.80186667 | −2.4638 | 89.14 | gravel and piece of biogenic (shells) sediment: gravel |
| 12/06/2010 | 5 | B31H1 | 31 | H1 | 49.80378333 | −2.4664 | 88.14 | gravel and piece of biogenic (shells) sediment: gravel |
| 12/06/2010 | 5 | B31H2 | 31 | H2 | 49.80378333 | −2.4664 | 88.14 | gravel and piece of biogenic (shells) sediment: gravel |
| 12/06/2010 | 5 | B32G | 32 | G | 49.8089 | −2.47075 | 93.14 | sand and gravel: sandy gravel |
| 12/06/2010 | 5 | B32H1 | 32 | H1 | 49.81075 | −2.47056667 | 96.14 | sand and gravel: sandy gravel |
| 12/06/2010 | 5 | ROV13 | 34 | 1 | 49.79965 | −2.22531667 | 61.14 | |
| 12/06/2010 | 5 | B36G | 36 | G | 49.80905 | −2.22628333 | 60.14 | clean gravel and pebbles: sandy gravel and pebbles |
| 12/06/2010 | 5 | B36H1 | 36 | H1 | 49.80718333 | −2.23335 | 67.64 | clean gravel and pebbles: sandy gravel and pebbles |
| 12/06/2010 | 5 | B36H2 | 36 | H2 | 49.80498333 | −2.23906667 | 66.14 | clean gravel and pebbles: sandy gravel and pebbles |
| 12/06/2010 | 5 | B37G | 37 | G | 49.81445 | −2.23598333 | 74.14 | coarse sand |
| 12/06/2010 | 5 | B38G | 38 | G | 49.816 | −2.23496667 | 82.14 | pebbles and sand: sandy gravel and pebbles |
| 12/06/2010 | 5 | B38H1 | 38 | H1 | 49.81566667 | −2.23761667 | 80.14 | pebbles and sand: sandy gravel and pebbles |
| 12/06/2010 | 5 | B38H2 | 38 | H2 | 49.81473333 | −2.24168333 | 80.14 | pebbles and sand: sandy gravel and pebbles |
| 12/06/2010 | 4 | B39G | 39 | G | 49.9168 | −1.92555 | 71.02 | pebbles + incrusting fauna |
| 12/06/2010 | 4 | ROV14 | 39 | 1 | 49.91705 | −1.92003333 | 71.3 | |
| 12/06/2010 | 4 | B41G | 41 | G | 49.88421667 | −1.77513333 | 74.74 | clean gravel: gravel |
| 12/06/2010 | 4 | B42G | 42 | G | 49.90035 | −1.708 | 68.13 | clean gravel and pebbles |

| DATE | Box | Code | ST | RE | LATITUDE | LONGITUDE | Depth | SEDIMENT PHOTOS |
|---|---|---|---|---|---|---|---|---|
| 13/06/2010 | 4 | ROV15 | 42 | 1 | 49.89766667 | −1.70868333 | 68.52 | |
| 13/06/2010 | 4 | ROV16 | 42 | 1 | 49.90033333 | −1.71645 | 68.96 | |
| 13/06/2010 | 3 | B43G | 43 | G | 49.98391667 | −1.2842 | 51.00 | gravel and coarse sand: sandy gravel |
| 13/06/2010 | 3 | B44G | 44 | G | 50.03398333 | −1.27103333 | 62.86 | gravel and coarse sand: sandy gravel |
| 13/06/2010 | 3 | B45G | 45 | G | 50.01706667 | −1.34053333 | 67.11 | pebbles, sand and gravel: sandy pebble + *Ophiothrix fragilis* |
| 13/06/2010 | 3 | B46G | 46 | G | 49.98328333 | −1.34286667 | 63.36 | pebbles, sand and gravel: sandy pebble |
| 13/06/2010 | 3 | B47G | 47 | G | 50.01678333 | −1.15681667 | 62.58 | pebbles, sand and gravel: sandy pebble |
| 13/06/2010 | 2 | B48G | 48 | G | 50.11686667 | −0.89631667 | 45.7 | pebbles and coarse sand: sandy pebbles + *Ophiothrix fragilis* |
| 13/06/2010 | 2 | B49G | 49 | G | 50.1181 | −0.84986667 | 44.9 | pebbles and coarse sand: sandy pebbles + *Ophiothrix fragilis* |
| 13/06/2010 | 2 | B50G | 50 | G | 50.1029 | −0.75853333 | 50.3 | coarse sand, gravel and shells: gravelly sand |
| 13/06/2010 | 2 | B51G | 51 | G | 50.10091667 | −0.68713333 | 46.2 | pebbles, gravel and coarse sand: sandy pebbles + *Ophiothrix fragilis* |
| 13/06/2010 | 2 | B52G | 52 | G | 50.08298333 | −0.8852 | 55.2 | gravel with low coarse sand: gravel |
| 13/06/2010 | 2 | ROV17 | 52 | 1 | 50.08451667 | −0.88268333 | 58.8 | |
| 13/06/2010 | 2 | ROV18 | 51 | 1 | 50.09803333 | −0.6792 | 53.8 | |
| 14/10/2010 | 1 | ROV19 | 53 | 1 | 50.14986667 | −0.35148333 | 46.8 | |
| 14/10/2010 | 1 | B53G | 53 | G | 50.14876667 | −0.35038333 | 43.4 | coarse sand, gravel: gravelly sand |
| 14/10/2010 | 1 | B54G | 54 | G | 50.16706667 | −0.25016667 | 42.64 | gravel, coarse sand, shells: sandy gravel |
| 14/10/2010 | 1 | B55G | 55 | G | 50.11571667 | −0.25716667 | 45.03 | gravel, coarse sand: sandy gravel |
| 14/10/2010 | 1 | B56G | 56 | G | 50.11535 | −0.3119 | 41.91 | coarse sand, gravel and shells: gravelly sand |
| 19/06/2011 | 7 | ROV101 | 101 | 1 | 49.09993333 | −2.73045 | 59.14 | |
| 20/06/2011 | 7 | B101G | 101 | G | 49.08051667 | −2.66063333 | 54.14 | pebbles, sand and gravel: sandy gravel and pebbles + *Ophiothrix fragilis* |
| 20/06/2011 | 7 | B102G | 102 | G | 49.08053333 | −2.69463333 | 55.14 | pebbles, sand and gravel: sandy gravel and pebbles + *Ophiothrix fragilis* |
| 20/06/2011 | 7 | B102H1 | 102 | H1 | 49.08188333 | −2.6992 | 54.14 | pebbles, sand and gravel: sandy gravel and pebbles + *Ophiothrix fragilis* |
| 20/06/2011 | 7 | B102H2 | 102 | H2 | 49.08021667 | −2.69173333 | 57.14 | pebbles, sand and gravel: sandy gravel and pebbles + *Ophiothrix fragilis* |
| 20/06/2011 | 7 | B103G | 103 | G | 49.09696667 | −2.68266667 | 56.14 | pebbles, sand and gravel: sandy gravel and pebbles + *Ophiothrix fragilis* |
| 20/06/2011 | 7 | B104G | 104 | G | 49.09885 | −2.65931667 | 56.14 | pebbles, sand and gravel: sandy gravel and pebbles |
| 20/06/2011 | 7 | B105H1 | 105 | H1 | 49.09933333 | −2.61485 | 56.14 | pebbles, sand and gravel: sandy gravel and pebbles + *Ophiothrix fragilis* |
| 20/06/2011 | 7 | B105H2 | 105 | H2 | 49.09958333 | −2.6116 | 56.14 | gravel and coarse sand: sandy gravel and pebbles + *Ophiothrix fragilis* |
| 20/06/2011 | 7 | B105G | 105 | G | 49.16633333 | −2.61546667 | 56.14 | pebbles and sand: sandy gravel and pebbles + *Ophiothrix fragilis* |
| 20/06/2011 | 7 | B106G | 106 | G | 49.1700 | −2.61546667 | 57.14 | pebbles and sand: sandy gravel and pebbles + *Ophiothrix fragilis* |
| 20/06/2011 | 8 | ROV102 | 102 | 1 | 48.99886667 | −3.5102 | 76.74 | |
| 20/06/2011 | 8 | ROV103 | 103 | 2 | 48.97805 | −3.60098333 | 76.14 | |
| 21/06/2011 | 8 | B107G | 107 | G | 48.96905 | −3.5337 | 73.14 | gravel and pebbles: sandy gravel and pebbles |
| 21/06/2011 | 8 | B108G | 108 | G | 48.95188333 | −3.57421667 | 73.14 | coarse sand, gravel: sandy gravel and pebbles |
| 21/06/2011 | 8 | B108H1 | 108 | H1 | 48.9519 | −3.57325 | 72.64 | coarse sand, gravel: gravelly sand |
| 21/06/2011 | 8 | B109H1 | 109 | H1 | 48.9685 | −3.61376667 | 76.14 | coarse sand, gravel with pebbles: gravelly sand with pebbles |
| 21/06/2011 | 8 | B109G | 109 | G | 48.96731667 | −3.61525 | 76.14 | coarse sand, gravel with pebbles: gravelly sand with pebbles |
| 21/06/2011 | 8 | B109H2 | 109 | H2 | 48.96863333 | −3.6084 | 76.14 | coarse sand, gravel with pebbles: gravelly sand with pebbles |
| 21/06/2011 | 8 | B110G | 110 | G | 48.98176667 | −3.58828333 | 76.14 | coarse sand, gravel with pebbles: gravelly sand with pebbles |
| 24/06/2011 | 9 | B112G | 112 | G | 48.78071667 | −3.98513333 | 67.14 | pebbles with incrusting fauna |
| 24/06/2011 | 9 | B113G | 113 | G | 48.81758333 | −3.95718333 | 74.14 | pebbles with incrusting fauna + *Ophiothrix fragilis* |
| 24/06/2011 | 9 | B114H1 | 114 | H1 | 48.83406667 | −3.94606667 | 77.14 | coarse sand, gravel with pebbles: gravelly sand with pebbles |
| 24/06/2011 | 9 | B114G | 114 | G | 48.83398333 | −3.94153333 | 77.14 | coarse sand, gravel with pebbles: gravelly sand with pebbles |
| 24/06/2011 | 9 | B114H2 | 114 | H2 | 48.834 | −3.93883333 | 77.14 | coarse sand and gravel: gravelly sand |
| 24/06/2011 | 9 | ROV104 | 116 | 1 | 48.80266667 | −4.00463333 | 83.14 | |
| 24/06/2011 | 9 | ROV105 | 111 | 1 | 48.78348333 | −3.9953 | 69.14 | |
| 24/06/2011 | 9 | B117G | 117 | G | 48.83215 | −4.07756667 | 83.14 | coarse sand with pebbles: gravelly sand and pebbles |

| DATE | Box | Code | ST | RE | LATITUDE | LONGITUDE | Depth | SEDIMENT PHOTOS |
|---|---|---|---|---|---|---|---|---|
| 25/06/2011 | 10 | B118G | 118 | G | 48.83671667 | −4.37348333 | 93.14 | coarse sand, gravel with pebbles: gravelly sand with pebbles |
| 25/06/2011 | 10 | B119H1 | 119 | H1 | 48.80138333 | −4.3521 | 89.14 | coarse sand with rare pebbles: sandy gravel and pebbles |
| 25/06/2011 | 10 | B119G | 119 | G | 48.80155 | −4.35165 | 89.14 | coarse sand with rare gravel: sandy gravel and pebbles |
| 25/06/2011 | 10 | B119H2 | 119 | H2 | 48.80155 | −4.351 | 89.14 | coarse sand, gravel with pebbles: gravelly sand with pebbles |
| 25/06/2011 | 10 | B120G | 120 | G | 48.80156667 | −4.38251667 | 89.14 | coarse sand, gravel with pebbles: gravelly sand with pebbles |
| 25/06/2011 | 10 | B120H1 | 120 | H1 | 48.80116667 | −4.38211667 | 90.14 | coarse sand, gravel with pebbles: gravelly sand with pebbles |
| 25/06/2011 | 10 | B120H2 | 120 | H2 | 48.80103333 | −4.38196667 | 90.14 | coarse sand, gravel with pebbles: gravelly sand with pebbles |
| 25/06/2011 | 10 | B121G | 121 | G | 48.8035 | −4.44663333 | 91.14 | coarse sand, gravel with pebbles: gravelly sand with pebbles |
| 25/06/2011 | 11 | B123G | 123 | G | 49.2176 | −3.47475 | 74.14 | coarse sand with pebbles: gravelly sand and pebbles |
| 26/06/2011 | 11 | B124G | 124 | G | 49.21748333 | −3.40815 | 71.14 | coarse sand with rare pebbles: gravelly sand and pebbles |
| 26/06/2011 | 11 | B125G | 125 | G | 49.21843333 | −3.38245 | 72.14 | coarse sand with rare pebbles: gravelly sand and pebbles |
| 26/06/2011 | 11 | B125H1 | 125 | H1 | 49.21878333 | −3.38706667 | 71.14 | pebbles and coarse sand: gravelly sand and pebbles |
| 26/06/2011 | 11 | B125H2 | 125 | H2 | 49.21883333 | −3.38893333 | 72.14 | pebbles and coarse sand: gravelly sand and pebbles |
| 26/06/2011 | 11 | B126H1 | 126 | H1 | 49.2001 | −3.38335 | 71.14 | coarse sand (biogenic) |
| 26/06/2011 | 11 | B126H2 | 126 | H2 | 49.2002 | −3.38675 | 71.14 | coarse sand (biogenic) |
| 26/06/2011 | 11 | B126G | 126 | G | 49.20013333 | −3.38831667 | 72.14 | coarse sand (biogenic) |
| 26/06/2011 | 11 | ROV108 | 126 | 1 | 49.20013333 | −3.37973333 | 72.14 | |
| 26/06/2011 | 11 | ROV109 | 128 | 1 | 49.18188333 | −3.41553333 | 73.14 | |
| 26/06/2011 | 11 | B128G | 128 | G | 49.18428333 | −3.4119 | 74.14 | coarse sand (biogenic) with pebbles |
| 26/06/2011 | 11 | B129H1 | 129 | H1 | 49.16468333 | −3.43968333 | 76.14 | coarse sand with pebbles |
| 26/06/2011 | 11 | B129G | 129 | G | 49.16508333 | −3.43923333 | 76.14 | coarse sand, gravel with pebbles: gravelly sand with pebbles |
| 27/06/2011 | 13 | ROV110 | 130 | 1 | 49.09433333 | −5.05886667 | 102.14 | |
| 27/06/2011 | 13 | ROV111 | 131 | 1 | 49.10693333 | −5.03501667 | 100.43 | |
| 27/06/2011 | 13 | B130H1 | 130 | H1 | 49.09991667 | −5.00065 | 100.18 | coarse sand, gravel with pebbles and shells: gravelly sand |
| 27/06/2011 | 13 | B130H1 | 130 | H2 | 49.09955 | −4.99801667 | 100.15 | coarse sand with gravel and shells: gravelly sand |
| 27/06/2011 | 13 | B130G | 130 | G | 49.09928333 | −4.99576667 | 100.11 | coarse sand with gravel and shells: gravelly sand |
| 27/06/2011 | 13 | B131H1 | 131 | H1 | 49.10596667 | −5.03701667 | 95.53 | coarse sand (biogenic) |
| 27/06/2011 | 13 | B131H2 | 131 | H2 | 49.10588333 | −5.0351 | 95.03 | coarse sand (biogenic) |
| 27/06/2011 | 13 | B131G | 131 | G | 49.10586667 | −5.03308333 | 96.03 | coarse sand (biogenic) |
| 27/06/2011 | 13 | B132G | 132 | G | 49.12751667 | −4.98456667 | 99.02 | coarse sand, gravel: gravelly sand |
| 27/06/2011 | 13 | B133H1 | 133 | H1 | 49.13601667 | −4.9648 | 96.04 | coarse sand (biogenic) |
| 27/06/2011 | 13 | B133H2 | 133 | H2 | 49.13605 | −4.96331667 | 100.04 | medium sand with pebbles |
| 27/06/2011 | 13 | B133G | 133 | G | 49.13615 | −4.9621 | 99.06 | medium sand with shells |
| 27/06/2011 | 13 | B134H1 | 134 | H1 | 49.1221 | −4.94716667 | 99.12 | coarse sand, gravel: gravelly sand |
| 27/06/2011 | 13 | B134H2 | 134 | H2 | 49.122 | −4.94568333 | 99.14 | gravel with coarse sand: sandy gravel |
| 27/06/2011 | 13 | B134G | 134 | G | 49.12195 | −4.9425 | 101.43 | coarse sand, gravel: gravelly sand |
| 27/06/2011 | 12 | ROV112 | 135 | 1 | 49.50126 | −3.95402 | 92.41 | |
| 27/06/2011 | 12 | ROV113 | 136 | 1 | 49.51450 | −3.942804 | 95.22 | |

## Appendix B

Granulometric composition in percentage of dry sediment of the 40 benthic stations (pebbles: >20 mm; large gravel: >5 mm; gravel: 2–5 mm; sand: 2 mm–63 μm; silt–clay: <63 μm) with Folk classification.

| Box | Station | Pebbles | Large Gravel | Gravel | Sand | Silt–Clay | Sediment Type |
|---|---|---|---|---|---|---|---|
| 1 | 2 | 0 | 37.21 | 11.66 | 49.24 | 1.89 | Sandy Gravel |
| | 3 | 0 | 28.83 | 13.54 | 56.37 | 1.27 | Sandy Gravel |
| | 4 | 0 | 59.07 | 7.04 | 32.73 | 1.16 | Sandy Gravel |
| | 5 | 0 | 67.84 | 8.30 | 22.81 | 1.06 | Sandy Gravel |
| 2 | 6 | 0 | 74.47 | 2.64 | 22.25 | 0.64 | Sandy Gravel |
| | 7 | 0 | 63.91 | 9.16 | 26.77 | 0.16 | Sandy Gravel |
| | 8 | 0 | 59.08 | 9.92 | 30.96 | 0.04 | Sandy Gravel |
| | 9 | 0 | 46.07 | 18.49 | 35.13 | 0.31 | Sandy Gravel |
| 3 | 10 | 0 | 0.32 | 54.77 | 44.91 | 0 | Sandy Gravel |
| | 11 | 0 | 81.12 | 6.58 | 12.13 | 0.17 | Gravel |
| | 12 | 0 | 76.21 | 9.75 | 13.55 | 0.49 | Gravel |
| | 14 | 0 | 77.13 | 8.58 | 13.50 | 0.79 | Gravel |
| | 15 | 0 | 81.99 | 5.88 | 12.05 | 0.08 | Gravel |
| | 16 | 0 | 62.88 | 6.14 | 30.87 | 0.11 | Sandy Gravel |
| 4 | 18 | 0 | 95.12 | 1.23 | 3.53 | 0.13 | Gravel |
| | 19 | 0 | 94.55 | 1.19 | 4.02 | 0.24 | Gravel |
| | 20 | 0 | 78.77 | 7.74 | 13.45 | 0.04 | Gravel |
| | 21 | 0 | 50.00 | 46.52 | 3.47 | 0.01 | Gravel |
| 5 | 31 | 0 | 91.66 | 5.27 | 2.95 | 0.12 | Gravel |
| | 32 | 0 | 67.04 | 8.03 | 24.50 | 0.43 | Sandy Gravel |
| | 36 | 0 | 83.65 | 1.19 | 15.13 | 0.02 | Gravel |
| | 38 | 0 | 84.78 | 4.64 | 10.58 | 0.01 | Gravel |
| 6 | 22 | 0 | 41.52 | 12.70 | 44.02 | 1.76 | Sandy Gravel |
| | 23 | 0 | 22.97 | 9.80 | 67.07 | 0.16 | Sandy Gravel |
| | 24 | 0 | 37.40 | 13.80 | 48.21 | 0.58 | Sandy Gravel |
| | 28 | 0 | 47.17 | 13.44 | 39.14 | 0.26 | Sandy Gravel |
| 7 | 102 | 16.64 | 22.77 | 9.80 | 50.35 | 0.44 | Sandy Gravel and Pebbles |
| | 105 | 53.60 | 10.67 | 8.67 | 26.58 | 0.48 | Sandy Gravel and Pebbles |
| 8 | 108 | 8.59 | 49.44 | 11.20 | 30.71 | 0.06 | Sandy Gravel and Pebbles |
| | 109 | 8.91 | 35.63 | 17.42 | 37.93 | 0.11 | Sandy Gravel and Pebbles |
| 9 | 114 | 40.15 | 18.62 | 9.99 | 31.21 | 0.02 | Sandy Gravel and Pebbles |
| 10 | 119 | 26.97 | 10.64 | 8.85 | 53.52 | 0.03 | Sandy Gravel and Pebbles |
| | 120 | 42.94 | 7.36 | 2.94 | 46.74 | 0.02 | Sandy Gravel and Pebbles |
| 11 | 125 | 21.54 | 6.73 | 13.74 | 57.98 | 0.02 | Sandy Gravel and Pebbles |
| | 126 | 0 | 1.92 | 17.16 | 80.89 | 0.03 | Gravelly Sand |
| | 129 | 37.05 | 7.15 | 5.24 | 50.52 | 0.03 | Sandy Gravel and Pebbles |
| | 130 | 17.97 | 10.48 | 12.30 | 58.84 | 0.41 | Sandy Gravel and Pebbles |

| Box | Station | Pebbles | Large Gravel | Gravel | Sand | Silt–Clay | Sediment Type |
|---|---|---|---|---|---|---|---|
|  | 131 | 0 | 4.92 | 15.65 | 79.35 | 0.07 | Gravelly Sand |
| 13 | 133 | 8.39 | 10.96 | 11.31 | 68.77 | 0.57 | Sandy Gravel and Pebbles |
|  | 134 | 28.40 | 17.55 | 8.96 | 44.65 | 0.45 | Sandy Gravel and Pebbles |

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
