# Peer review of "A Multidisciplinary Approach for A Better Knowledge of the Benthic Habitat and Community Distribution in the Central and Western English Channel"

_jmse, doi:10.3390/jmse10081112_

Round 1

Reviewer 1 Report

Review

Paper title: A Multidisciplinary Approach for a Better Knowledge of Benthic Habitat and Community Distribution in the Central and Western English Channel

 The authors provided an analysis of sediment and benthic data collected in the Central and Western English Channel and presented a more suitable classification of benthic habitats using combined datasets of classical data obtained by grab as well as sonar and video data. These findings may have important implications for monitoring benthic communities in the study area.

 All these reasons explain the relevance of the paper by Jean-Claude Dauvin and co-authors submitted to "JMSE".

 General scores.

 The data presented by the authors are original and significant. The study is correctly designed and the authors used appropriate sampling methods. In general, statistical analyses are performed with good technical standards. The authors conducted careful work that may attract the attention of a wide range of specialists focused on benthic ecology.

Major concerns.

The abstract should be shortened.

The authors should provide a caption for each table and figure.

L 212. Please, clarify how you separate “common” and ‘abundant” taxa.

L 250-259. This section rather corresponds to “Materials and Methods”

L 352-354. Please, re-write. It is difficult to understand.

In general, the English should be improved.

Specific comments.

L 21. Change “to use grab corers is” to “of using grab corers in”

L 22. Change “had been” to “was”

L 29. Change “had been” to “were”

L 31. Change “Three others” to “Three other”

L 32. Change “associated to” to “associated with”

L 33. Change “bristle star” to “the bristle star”

L 33. “Ophiothrix fragilis” should be italicized.

L 34. Change “magnitude with” to “magnitude as the”

L 37. Change “determining to” to “determining by”

L 41. Change “it was suggested” to “it is suggested”

L 42. Change “on a 1 mm mesh for all the area” to “in a 1 mm mesh for the entire area”

L 44. Change “was also” to “is also”

L 49. Change “fundamental” to “a fundamental”

L 50. Change “carried out” to “determined”

L 54. Change “had been carried out across the three phases this” to “has been carried out across the three phases of this”

L 57. Change “technic” to “technique”

L 57. Change “dredge for” to “dredge by”

L 61. Change “hydrodynamic on” to “hydrodynamic in”

L 63. Change “zone with high currents which can arise up to 5 knots in some area” to “the zone with high currents which can arise up to 5 knots in some areas”

L 67. Change “input” to “the input”

L 68. Change “The results are” to “The result is”

L 74. Change “difficulties to sample” to “difficulties of sampling

L 76. Change “collected in near” to “collected near”

L 84. Change “[16] Used” to “[16] used”

L 116. Change “[11] had completed” to “[11] completed”

L 123. Change “have been” to “was”

L 127. Change “south to” to “south of”

L 127. Change “The boxes” to “Boxes”

L 130. Change “had been” to “were”

L 144. Change “surveys 2010” to “surveys in 2010”

L 144. Change “methodology” to “the methodology”

L 158. Change “Data was” to “Data were”

L 169. Change “benthic” to “the benthic”

L 191. Change “except the Box 10” to “except Box 10”

L 192. Change “video” to “of video”

L 198. Change “used identify” to “used to identify”

L 200. Change “numbers” to “number”

L 205. Change “according” to “according to”

L 206. Change “incrusting” to “encrusting”

L 207-211. The Latin names should be italicized here and throughout the remaining text.

L 218. Change “case” to “cases”

L 219. Change “apart” to “apart from”

L 245. Change “The figure 2 showed” to “Figure 2 shows”

L 247. Change “at the box 7 and 84.5 km at the” to “in box 7 and 84.5 km in”

L 260. Change “The figure 3 gave” to “Figure 3 gives”

L 275. Change “been seen” to “be seen”

L 278. Change “the box 3” to “box 3”

L 282. Change “Brittany” to “the Brittany”

L 295. Change “classed” to “classified”

L 298. Change “Brittany” to “the Brittany”

L 299. Change “classified in” to “classified into”

L 302. Change “gradient” to “gradients”

L 316. Change “was recorded” to “were recorded”

L 319. Delete “accounted” 

L 324. Change “An additional” to “Additional”

L 327. Change “recorded” to “recorded in”

L 335. Change “at the station” to “at station”

L 335. Change “at the station” to “at station”

L 336. Change “at the station” to “at station”

L 337. Change “at the station” to “at station”

L 338. Change “to to” to “to”

L 340. Change “the box” to “box”

L 341. Change “the box” to “box”

L 342. Change “the boxes” to “boxes”

L 342. Change “the boxes” to “boxes”

L 344. Change “at the station” to “at station”

L 345. Change “at the station” to “at station”

L 345. Change “at the station” to “at station”

L 346. Change “at the station” to “at station”

L 348. Change “at the bow 8 to 261 at the” to “at box 8 to 261 at”

L 350. Change “included between 100 and 200 200” to “between 100 and 200”

L 356. Change “On the 40” to “Of the 40”

L 360. Change “On the 40” to “Of the 40”

L 371. Change “the  Box” to “Box”

L 376. Change “The group” to “Group”

L 376. Change “the boxes” to “boxes”

L 377. Change “The group” to “Group”

L 381. Change “the boxes” to “boxes”

L 393. Change “The group a was similar to the group” to “Group a was similar to group”

L 394. Change “the box” to “box”

L 395. Change “The group” to “Group”

L 398. Change “The group” to “Group”

L 401. Change “The group” to “Group”

L 409. Change “was  considered” to “were  considered”

L 411. Change “The group” to “Group”

L 412. Change “previous analyses” to “the previous analyses”

L 414. Change “The group” to “Group”

L 415. Change “analysis” to “analyses”

L 415. Change “the station” to “station”

L 418. Change “The group” to “Group”

L 418. Change “the groups” to “groups”

L 419. Change “previous analyses” to “the previous analyses”

L 420. Change “The group” to “Group”

L 421. Change “the boxes” to “boxes”

L 424. Change “the group” to “group”

L 424. Change “the boxes” to “boxes”

L 428. Change “had  been  identified” to “were  identified”

L 431. Change “a level of 39%” to “a 39% level of”

L 431. Change “The station” to “Station”

L 433. Change “Both stations” to “Two stations”

L 434. Change “The group” to “Group”

L 436. Change “The group” to “Group”

L 438. Change “The group” to “Group”

L 440. Change “the group” to “group”

L 456. Change “(Raoux et al., 2021)” to “[51] or [59]”

L 458. Change “had been organised” to “were organised”

L 461. Change “are dominant” to “is dominant”

L 466. Change “at the” to “at”

L 474. Change “2000’s” to “2000s”

L 483. Change “the level” to “levels”

L 485. Change “classified in” to “classified into”

L 486. Change “changed of cluster group” to “changed cluster groups”

L 494. Change “both boxes” to “boxes”

L 496. Change “Both stations” to “Two stations”

L 498. Change “the level” to “level”

L 504. Change “matched to” to “matched”

L 514. Change “a large biomass” to “large biomass”

L 519. Change “into such detail” to “in such detail”

L 539. Change “individual” to “individuals”

L 545. Change “abundances included” to “abundances”

L 546. Change “the central part” to “in the central part”

L 547. Change “type” to “types”

L 547. Change “depth” to “depths”

L 553. Change “magnitude of” to “magnitude as”

L 560. Change “Difference of information’s between the seven descriptors on the benthic habitat” to “Difference in information between the seven descriptors of the benthic habitat”

L 571. Change “study of a” to “study from the”

L 574. Change “have need a too high sampling effort” to “need too high sampling effort”

L 576. Change “to lacks” to “to the lack”

L 578. Change “variability on” to “variability of”

L 579. Change “can be apply on” to “can be applied to”

L 583. Change “a  coding” to “coding”

L 584. Change “had suggested to add at the” to “suggested to add at”

L 598. Change “had shown” to “have shown”

L 609. Delete “had”

L 615. Change “spatial” to “spatially”

L 621. Change “the those” to “those”

L 624. Change “a good” to “good”

L 628. Change “component” to “components”

L 632. Change “had used” to “used”

L 644. Change “most” to “in most”

L 645. Change “had frequently” to “has been frequently”

L 650. Change “all the area” to “the entire area”

Author Response

Caen on 21 July 2022

Dear Editor

First at all, thanks for your evaluation and also thanks to both reviewers for their very useful comments and suggestions to improve our typescript.

You can find below in red our responses to the reviewers.

We hope that this version should be published in JMSE

Yours

Pr Jean-Claude Dauvin

Open Review

English language and style

( ) Extensive editing of English language and style required
(x) Moderate English changes required
( ) English language and style are fine/minor spell check required
( ) I don't feel qualified to judge about the English language and style

Yes

Can be improved

Must be improved

Not applicable

Does the introduction provide sufficient background and include all relevant references?

(x)

( )

( )

( )

Are all the cited references relevant to the research?

(x)

( )

( )

( )

Is the research design appropriate?

(x)

( )

( )

( )

Are the methods adequately described?

( )

(x)

( )

( )

Are the results clearly presented?

( )

( )

(x)

( )

Are the conclusions supported by the results?

(x)

( )

( )

( )

Comments and Suggestions for Authors

Review

Paper title: A Multidisciplinary Approach for a Better Knowledge of Benthic Habitat and Community Distribution in the Central and Western English Channel

The authors provided an analysis of sediment and benthic data collected in the Central and Western English Channel and presented a more suitable classification of benthic habitats using combined datasets of classical data obtained by grab as well as sonar and video data. These findings may have important implications for monitoring benthic communities in the study area.

All these reasons explain the relevance of the paper by Jean-Claude Dauvin and co-authors submitted to "JMSE".

General scores.

The data presented by the authors are original and significant. The study is correctly designed and the authors used appropriate sampling methods. In general, statistical analyses are performed with good technical standards. The authors conducted careful work that may attract the attention of a wide range of specialists focused on benthic ecology.

Major concerns.

The abstract should be shortened. This was made.

The authors should provide a caption for each table and figure. This is available at the end of the text.

L 212. Please, clarify how you separate “common” and ‘abundant” taxa. OK

L 250-259. This section rather corresponds to “Materials and Methods” OK, this was changed.

L 352-354. Please, re-write. It is difficult to understand. This was made.

In general, the English should be improved. Sophie Lozach who is a researcher of the CEFAS (UK) has checked again the text for the English style.

Specific comments.

L 21. Change “to use grab corers is” to “of using grab corers in” OK

L 22. Change “had been” to “was” OK

L 29. Change “had been” to “were” OK

L 31. Change “Three others” to “Three other” OK

L 32. Change “associated to” to “associated with” OK

L 33. Change “bristle star” to “the bristle star” OK

L 33. “Ophiothrix fragilis” should be italicized. OK

L 34. Change “magnitude with” to “magnitude as the” OK

L 37. Change “determining to” to “determining by”OK

L 41. Change “it was suggested” to “it is suggested” OK

L 42. Change “on a 1 mm mesh for all the area” to “in a 1 mm mesh for the entire area” OK

L 44. Change “was also” to “is also” OK

L 49. Change “fundamental” to “a fundamental” OK

L 50. Change “carried out” to “determined” OK

L 54. Change “had been carried out across the three phases this” to “has been carried out across the three phases of this” OK

L 57. Change “technic” to “technique” OK

L 57. Change “dredge for” to “dredge by” OK

L 61. Change “hydrodynamic on” to “hydrodynamic in” OK

L 63. Change “zone with high currents which can arise up to 5 knots in some area” to “the zone with high currents which can arise up to 5 knots in some areas” OK

L 67. Change “input” to “the input” OK

L 68. Change “The results are” to “The result is” OK

L 74. Change “difficulties to sample” to “difficulties of sampling” OK

L 76. Change “collected in near” to “collected near” OK

L 84. Change “[16] Used” to “[16] used” OK

L 116. Change “[11] had completed” to “[11] completed” OK

L 123. Change “have been” to “was” OK

L 127. Change “south to” to “south of” OK

L 127. Change “The boxes” to “Boxes” OK

L 130. Change “had been” to “were” OK

L 144. Change “surveys 2010” to “surveys in 2010” OK

L 144. Change “methodology” to “the methodology” OK

L 158. Change “Data was” to “Data were” OK

L 169. Change “benthic” to “the benthic” OK

L 191. Change “except the Box 10” to “except Box 10” OK

L 192. Change “video” to “of video” OK

L 198. Change “used identify” to “used to identify” OK

L 200. Change “numbers” to “number” OK

L 205. Change “according” to “according to” OK

L 206. Change “incrusting” to “encrusting” OK

L 207-211. The Latin names should be italicized here and throughout the remaining text. This was made in the first version the typescript but transform in roman letter in the pdf

L 218. Change “case” to “cases” OK

L 219. Change “apart” to “apart from” OK

L 245. Change “The figure 2 showed” to “Figure 2 shows” OK

L 247. Change “at the box 7 and 84.5 km at the” to “in box 7 and 84.5 km in” OK

L 260. Change “The figure 3 gave” to “Figure 3 gives” OK

L 275. Change “been seen” to “be seen” OK

L 278. Change “the box 3” to “box 3” OK

L 282. Change “Brittany” to “the Brittany” OK

L 295. Change “classed” to “classified” OK

L 298. Change “Brittany” to “the Brittany” OK

L 299. Change “classified in” to “classified into” OK

L 302. Change “gradient” to “gradients” OK

L 316. Change “was recorded” to “were recorded” OK

L 319. Delete “accounted” OK

L 324. Change “An additional” to “Additional” OK

L 327. Change “recorded” to “recorded in” OK

L 335. Change “at the station” to “at station” OK

L 335. Change “at the station” to “at station” OK

L 336. Change “at the station” to “at station” OK

L 337. Change “at the station” to “at station” OK

L 338. Change “to to” to “to” OK

L 340. Change “the box” to “box” OK

L 341. Change “the box” to “box” OK

L 342. Change “the boxes” to “boxes” OK

L 342. Change “the boxes” to “boxes” OK

L 344. Change “at the station” to “at station” OK

L 345. Change “at the station” to “at station” OK

L 345. Change “at the station” to “at station” OK

L 346. Change “at the station” to “at station” OK

L 348. Change “at the bow 8 to 261 at the” to “at box 8 to 261 at” OK

L 350. Change “included between 100 and 200 200” to “between 100 and 200” OK

L 356. Change “On the 40” to “Of the 40” OK

L 360. Change “On the 40” to “Of the 40” OK

L 371. Change “the  Box” to “Box” OK

L 376. Change “The group” to “Group” OK

L 376. Change “the boxes” to “boxes” OK

L 377. Change “The group” to “Group” OK

L 381. Change “the boxes” to “boxes” OK

L 393. Change “The group a was similar to the group” to “Group a was similar to group” OK

L 394. Change “the box” to “box” OK

L 395. Change “The group” to “Group” OK

L 398. Change “The group” to “Group” OK

L 401. Change “The group” to “Group” OK

L 409. Change “was  considered” to “were  considered” OK

L 411. Change “The group” to “Group” OK

L 412. Change “previous analyses” to “the previous analyses” OK

L 414. Change “The group” to “Group” OK

L 415. Change “analysis” to “analyses” OK

L 415. Change “the station” to “station” OK

L 418. Change “The group” to “Group” OK

L 418. Change “the groups” to “groups” OK

L 419. Change “previous analyses” to “the previous analyses” OK

L 420. Change “The group” to “Group” OK

L 421. Change “the boxes” to “boxes” OK

L 424. Change “the group” to “group” OK

L 424. Change “the boxes” to “boxes” OK

L 428. Change “had  been  identified” to “were  identified” OK

L 431. Change “a level of 39%” to “a 39% level of” OK

L 431. Change “The station” to “Station” OK

L 433. Change “Both stations” to “Two stations” OK

L 434. Change “The group” to “Group” OK

L 436. Change “The group” to “Group” OK

L 438. Change “The group” to “Group” OK

L 440. Change “the group” to “group” OK

L 456. Change “(Raoux et al., 2021)” to “[51] or [59]” OK

L 458. Change “had been organised” to “were organised” OK

L 461. Change “are dominant” to “is dominant”         OK

L 466. Change “at the” to “at” OK

L 474. Change “2000’s” to “2000s” OK

L 483. Change “the level” to “levels” OK

L 485. Change “classified in” to “classified into” OK

L 486. Change “changed of cluster group” to “changed cluster groups” OK

L 494. Change “both boxes” to “boxes” OK

L 496. Change “Both stations” to “Two stations” OK

L 498. Change “the level” to “level” OK

L 504. Change “matched to” to “matched” OK

L 514. Change “a large biomass” to “large biomass” OK

L 519. Change “into such detail” to “in such detail” OK

L 539. Change “individual” to “individuals”       OK

L 545. Change “abundances included” to “abundances” OK

L 546. Change “the central part” to “in the central part” OK

L 547. Change “type” to “types” OK

L 547. Change “depth” to “depths” OK

L 553. Change “magnitude of” to “magnitude as” OK

L 560. Change “Difference of information’s between the seven descriptors on the benthic habitat” to “Difference in information between the seven descriptors of the benthic habitat” OK

L 571. Change “study of a” to “study from the” OK

L 574. Change “have need a too high sampling effort” to “need too high sampling effort” OK

L 576. Change “to lacks” to “to the lack” OK

L 578. Change “variability on” to “variability of” OK

L 579. Change “can be apply on” to “can be applied to” OK

L 583. Change “a  coding” to “coding” OK

L 584. Change “had suggested to add at the” to “suggested to add at” OK

L 598. Change “had shown” to “have shown” OK

L 609. Delete “had” OK

L 615. Change “spatial” to “spatially” OK

L 621. Change “the those” to “those” OK

L 624. Change “a good” to “good”       OK

L 628. Change “component” to “components” OK

L 632. Change “had used” to “used” OK

L 644. Change “most” to “in most” OK

L 645. Change “had frequently” to “has been frequently” OK

L 650. Change “all the area” to “the entire area” OK

Submission Date

01 June 2022

Date of this review

05 Jun 2022 10:21:22

Reviewer 2 Report

General remarks:

The paper deals with an important topic regarding the structure and functions of benthic habitats in EC, the area that still has many gaps in the knowledge of the distribution of these habitats, due to poor accessibility in its investigation. The work is all the more important as extensive economic developments are expected in the area by installing offshore wind farms. On the other hand, there are not many such interdisciplinary studies, and the existing ones are not enough for the predictive models that map benthic habitats. Then the existence of a decreasing number of taxonomists raises the value of the data obtained in this study, because currently most are oriented towards methods of rapid determination of species, such as eDNA, and morphological determinations are increasingly rare.

The results displayed in the paper is a good example of accuracy of applied methodological principles with high confidence at fine scale resolution which can feed and serve for the European Marine Data portals. The paper focused on methodological approaches of benthic habitat mapping with respect of enforced and accepted rules. All described issues follow the modern usage of remote sensing techniques which is mandatory in this field of activity in order to quantify the habitat extent.

I propose the publication of this paper after completing / correcting some minor leaks, such as:

-          Italization of the name of the species in the whole manuscript.

-          It is preferable for each table and figure to have a title and legend, so that it is easier for its reader to understand (for example, in table 2 the colored stations appear, but it is not mentioned which color each ES corresponds to).

-          228 line: it is mentioned “Ecological Status was estimated from H’ and J values according to the thresholds defined previously (Pezy and Dauvin, 2021). I may not be right, but in the paper mentioned the threshold values are used after H '- Vincent et al. (2002), J - Dauvin et al. (2017). Maybe it would be right to use these quotes.

-          The paper does not explain why only the H 'and J indices were chosen and why the others proposed by MSFD (AMBI, M-AMBI) or others were not used.

-          291; 363; 388; 408; 427; 449; 462 lines – delete the point in front of the subchapters.

-          The small similarity (24%) may indicate the existence of several subtypes of habitats, which sometimes overlap. However, these highlighted groups would be desirable to be mentioned / described in the legend of the figures, so that the reader can more easily associate the information.

-          577 line -  “…the comparison of sonar responses with EC sedimentary cover available at SHOM...” you could explain why you only used SHOM and not EuSeamap Seabeads habitats product.

-          The use of the sentence “… seven descriptors on the benthic habitat…” can induce the reader to think of the descriptors from MSFD, maybe it would be better “… seven methods…”.

Author Response

Open Review

English language and style

( ) Extensive editing of English language and style required
( ) Moderate English changes required
( ) English language and style are fine/minor spell check required
(x) I don't feel qualified to judge about the English language and style

Yes

Can be improved

Must be improved

Not applicable

Does the introduction provide sufficient background and include all relevant references?

(x)

( )

( )

( )

Are all the cited references relevant to the research?

(x)

( )

( )

( )

Is the research design appropriate?

(x)

( )

( )

( )

Are the methods adequately described?

(x)

( )

( )

( )

Are the results clearly presented?

(x)

( )

( )

( )

Are the conclusions supported by the results?

(x)

( )

( )

( )

Comments and Suggestions for Authors

General remarks:

The paper deals with an important topic regarding the structure and functions of benthic habitats in EC, the area that still has many gaps in the knowledge of the distribution of these habitats, due to poor accessibility in its investigation. The work is all the more important as extensive economic developments are expected in the area by installing offshore wind farms. On the other hand, there are not many such interdisciplinary studies, and the existing ones are not enough for the predictive models that map benthic habitats. Then the existence of a decreasing number of taxonomists raises the value of the data obtained in this study, because currently most are oriented towards methods of rapid determination of species, such as eDNA, and morphological determinations are increasingly rare.

The results displayed in the paper is a good example of accuracy of applied methodological principles with high confidence at fine scale resolution which can feed and serve for the European Marine Data portals. The paper focused on methodological approaches of benthic habitat mapping with respect of enforced and accepted rules. All described issues follow the modern usage of remote sensing techniques which is mandatory in this field of activity in order to quantify the habitat extent.

I propose the publication of this paper after completing / correcting some minor leaks, such as:

-          Italization of the name of the species in the whole manuscript. OK This was made in the first version the typescript but transform in roman letter in the pdf

-          It is preferable for each table and figure to have a title and legend, so that it is easier for its reader to understand (for example, in table 2 the colored stations appear, but it is not mentioned which color each ES corresponds to). This was included in the legend of the table 2 The colour coding corresponds to the Ecological Status of the Water Framework Directive: blue, high status; green, good status; yellow, moderate status; orange, poor status, and red bad status.

-          228 line: it is mentioned “Ecological Status was estimated from H’ and J values according to the thresholds defined previously (Pezy and Dauvin, 2021). I may not be right, but in the paper mentioned the threshold values are used after H '- Vincent et al. (2002), J - Dauvin et al. (2017). Maybe it would be right to use these quotes. Resumed was added in the text, t is true that original threshold was defined before by these authors.

-          The paper does not explain why only the H 'and J indices were chosen and why the others proposed by MSFD (AMBI, M-AMBI) or others were not used. These indices were proposed to identify the response of benthic macrofauna to an increase of organic matter and were not adapted for coarse sand sediment with very low concentration of OM. This was added in the texte (lines 236 and 239).

-          291; 363; 388; 408; 427; 449; 462 lines – delete the point in front of the subchapters. These points do not appeared in the original version?

-          The small similarity (24%) may indicate the existence of several subtypes of habitats, which sometimes overlap. However, these highlighted groups would be desirable to be mentioned / described in the legend of the figures, so that the reader can more easily associate the information. This level of similarity was chosen to give interpretable assemblages.

-          577 line -  “…the comparison of sonar responses with EC sedimentary cover available at SHOM...” you could explain why you only used SHOM and not EuSeamap Seabeads habitats product. French SHOM products was very efficient for the surficial sediment type in the English Channel; efficient was added in the text. But we have taken into account this suggestion and add a figure (10) in our paper (EUNIS habitat map from the library of marine habitats maps in European waters (EMODnet seabed habitats). Videocharm side scan sonar survey lines are showed with corresponding box number) and add the following comments. Figure 10 shows EUNIS habitat map from EUSeaMap product (https://www.emodnet-seabedhabitats.eu) with the VIDEOCHARM side scan sonar survey lines over imposed. VIDEOCHARM data sets confirm our EUNIS classification for boxes 1 and 2. Box 11 cover partially both a high and a moderate energy circalittoral areas according EUSeaMap. The difference between these two areas is highlighted on the side scan sonar profile with presence of mega dunes on the highest energy part (the two northwest profiles of the box 11). This is also the case for the box 13, entirely located on a high energy circalittoral rock area, which shows mega dunes both with sonar profile data and ROV. Moreover, it will be interesting in the future to integrate our VIDEOCHARM data in the EUSeaMap.

-          The use of the sentence “… seven descriptors on the benthic habitat…” can induce the reader to think of the descriptors from MSFD, maybe it would be better “… seven methods…”. This was made.

Submission Date

01 June 2022

Date of this review

10 Jul 2022 15:01:40

Round 2

Reviewer 1 Report

Second review

Paper title: A Multidisciplinary Approach for a Better Knowledge of Benthic Habitat and Community Distribution in the Central and Western English Channel

The authors improved the paper but one concern was not fixed.

L 222. The authors stated that: "Three classes of abundance of the taxa were established: 1) rare,

one to some individuals, 2) common taxa and 3) abundant taxa".

Please, clarify how you separate “common” and "abundant” taxa. Provide the criteria for common and abundant taxa as you did for rare taxa.

L 22. Change “is such hard” to “in such hard”

L 40. Change “not determining” to “not determined”

L 53. Change “this” to “of this”

L 66. Change “some part” to “some parts”

L 152. Change “either” to “on either”

L 227. Change “The  Appendix” to “Appendix”

L 231. Change “only one grab were” to “only one grab was”

L 265. Change “The  figure” to “Figure”

L 271. Change “a rugged” to “rugged”

L 332. Change “a 2 mm” to “2 mm”

L 375. Change “last group g” to “group g”

L 387. Change “Group a was similar to the group a” to “Group a from our study was similar to group a”

L 408. Change “the station 108” to “station 108”

L 411. Change “the group’s g and b” to “groups g and b”

L 479. Change “changed of” to “changed”

L 490. Change “very low number” to “a very low number”

L 579. Change “Box 11 cover partially both a” to “Box 11 covers partially both”

L 580. Change “according” to “according to”

L 582. Change “the box 11). This is also the case for the box 13” to “box 11). This is also the case for box 13”

L 597. Change “insight of” to “insight into”

L 648. Change “improvements using a 1-mm mesh sieve mesh is” to “improvements of using a 1-mm mesh sieve mesh are”

Author Response

Dear Editor

Thanks to the reviewer 2 for his second review on our paper and last to improve our typescript.

You can find below in red our responses to the reviewers.

We hope that this version should be published in JMSE

Yours

Pr Jean-Claude Dauvin
